# MR.PEA: The Meta-Reasoning Prompt Engineering Agent

## Abstract

Prompt optimization is critical for maximizing the performance of large language models (LLMs). However, it often relies on costly labeled data. Self-supervised methods reduce data dependency, but they suffer from optimization ambiguity or high computational costs. To address these limitations, we propose the Meta-Reasoning Prompt Engineering Agent (MR.PEA), a self-supervised prompt optimization framework that operates with minimal input. MR.PEA leverages meta-reasoning to iteratively build task-specific knowledge, including problem-solving strategies and evaluation criteria, while adaptively retrieving external information to enhance its understanding. This knowledge guides the generation of diverse validation examples, targeted prompt refinement, and comprehensive quality assessments. Experiments on GSM8K and Big-Bench Hard show that MR.PEA outperforms existing baselines, achieving an average performance gain of 7.4% with an optimization cost as low as $0.01 per task.

## 1 Introduction

The rapid advancement of large language models (LLMs) has revolutionized natural language processing, enabling complex reasoning and generation across diverse domains (Brown et al., 2020; Wei et al., 2022; OpenAI, 2024). Prompts, which serve as input instructions, are crucial for guiding LLMs to produce desired outcomes on specific tasks (Schulhoff et al., 2025). Effective prompting can even enable smaller models to match or surpass larger models (Belcak et al., 2025; Zhang et al., 2025a; Gao et al., 2025). However, LLMs are highly sensitive to prompt design. Minor changes in wording (Zhuo et al., 2024), instruction order (Chen et al., 2024), or formatting (Sclar et al., 2024; He et al., 2024; Voronov et al., 2024; Ngweta et al., 2025) can lead to performance variations. Manual prompt crafting (Kojima et al., 2022; Zhou et al., 2023a; Yao et al., 2023; Madaan et al., 2023; Zheng et al., 2024) is effective, but requires domain expertise and iterative refinement through trial-and-error. As LLM applications expand, the time and effort required for manual prompt engineering become unsustainable.

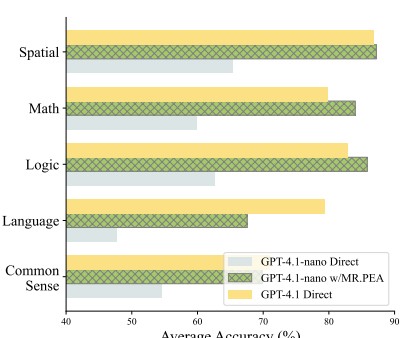

Figure 1: Performance comparison of GPT-4.1-nano Direct, GPT-4.1-nano with MR.PEA, and GPT-4.1 Direct on five BBH categories.

Automatic prompt optimization methods leverage LLMs to refine prompts and improve their performance on specific tasks. Among these, supervised methods (Pryzant et al., 2023; Zhou et al., 2023b; Guo et al., 2024; Fernando et al., 2024; Yang et al., 2024; Wang et al., 2024; Opsahl-Ong et al., 2024; Khattab et al., 2024; Ye et al., 2024; Tang et al., 2025; Zhang et al., 2025b) are particularly prominent due to their clear optimization signals. In each iteration, newly refined prompt candidates are tested on a validation set, and their performance is scored against ground truth using metrics like accuracy. The best prompts are then selected for further improvement in subsequent iterations. Yet, these methods typically require tens to hundreds of labeled examples for validation, which limits its scalability and applicability in low-resource scenarios.

Figure 2: Comparison of optimized prompts on BBH *sports_understanding*. The upper box shows minimal input. MR.PEA applies task-specific instructions, while SPO uses general reasoning.

Self-supervised prompt optimization eliminates the reliance on labeled data by leveraging the LLM's intrinsic knowledge, data patterns, and task information. This approach holds great promise for reducing the manual effort involved in prompt engineering. However, the absence of ground-truth answers poses a key challenge: *How can prompts be effectively and efficiently refined without explicit optimization signals?* This question underscores two critical issues of existing self-supervised methods: optimization ambiguity and high computational costs.

Optimization ambiguity arises from the lack of clear optimization signals, making it difficult to reliably assess prompt quality and continuously improve prompts. For instance, synthetic supervision methods (Agarwal et al., 2025) generate pseudo-labeled datasets using LLMs, but errors in these datasets can propagate through the optimization process, leading to misdirected refinements. Similarly, LLM-as-a-judge approaches (Xiang et al., 2025) optimize prompts based on user-defined task requirements. While promising, these methods rely heavily on the clarity of task specifications. Poorly defined criteria can result in overly general prompts that fail to capture task-specific nuances needed in complex tasks. Figure 2 provides an example of this issue.

High computational costs further limit the scalability of self-supervised methods. Consistency-based approaches (Zhang et al., 2024) attempt to improve reliability by analyzing the consistency of output answers across multiple runs and comparing answers from different prompts for adjustment. The adjusted consistency score is used as the optimization signal. However, this process is computation-intensive and can still fail to detect consistently incorrect outputs, leading to flawed optimization.

To tackle these challenges, we propose the Meta-Reasoning Prompt Engineering Agent (MR.PEA), a self-supervised framework that refines prompts with task-specific guidance under extreme data scarcity. MR.PEA introduces a meta-reasoning-driven optimization cycle that iteratively builds and refines task-specific knowledge. This knowledge includes (1) problem-solving strategies, which are generalizable techniques tailored to address task-specific nuances, and (2) reasoning-oriented evaluation criteria, which go beyond simple metrics to assess clarity, coherence, and logical consistency.

MR.PEA operates under minimal input requirements: a brief task description, a single unlabeled example, and a format specification. The optimization process begins with meta-reasoning about task-specific knowledge, where MR.PEA self-assesses its existing knowledge and dynamically updates it through internal reasoning or external information retrieval (e.g., web search). This curated knowledge directly guides the refinement process, ensuring that prompts are tailored to meet the unique requirements of each task. To further enhance robustness, MR.PEA generates diverse validation examples that vary in difficulty, context, and structure, mitigating the risk of overfitting to a single example. In summary, our contributions are:

▷ We propose MR.PEA, the Meta-Reasoning Prompt Engineering Agent, a self-supervised prompt optimization framework that operates with minimal input, requiring only a task description, a single unlabeled example, and format specification (Input example in Figure 2). This significantly reduces data dependency and manual effort.

▷ MR.PEA introduces a meta-reasoning-driven optimization cycle that adaptively builds and refines task-specific knowledge, including problem-solving strategies and reasoning-oriented evaluation criteria. This approach resolves optimization ambiguity by incorporating task-specific refinements and

quality assessments. MR.PEA also generates diverse validation examples to prevent overfitting to a single example and improve adaptability to complex tasks.

▷ Extensive experiments on GSM8k and Big-Bench Hard (22 task types) show that MR.PEA improves performance by an average of 7.4% and up to 18.7% on BBH math tasks in zero-shot strict parsing settings. It achieves this at a cost of \$0.01 per optimization per task. Prompts optimized by MR.PEA enable smaller LLMs to match or even surpass larger LLMs in direct invocation (Figure 1).

## 2 RELATED WORK

**Supervised Prompt Optimization.**   Supervised prompt optimization uses labeled datasets to refine prompts by evaluating their performance against ground truth metrics, such as accuracy. Most methods treat prompt optimization as a search or generation problem, where LLMs act as optimizers to generate new prompt candidates based on previous results. Some approaches use direct scoring (Zhou et al., 2023b), prompt guidance (Yang et al., 2024), or textual feedback (Pryzant et al., 2023; Wang et al., 2024; Ye et al., 2024) to select and improve prompts. Evolutionary algorithms (Guo et al., 2024; Fernando et al., 2024) treat prompts as a population that evolve through mutation and recombination. Recent work introduces gradient-inspired strategies (Tang et al., 2025), system-user prompt co-optimization (Zhang et al., 2025b), and few-shot learning (Opsahl-Ong et al., 2024; Khattab et al., 2024) to further boost performance. Despite their effectiveness, these methods depend on large amounts of high-quality labeled data. This dependency makes them unsuitable for tasks where labeled data is scarce, expensive, or unavailable.

**Self-Supervised Prompt Optimization.**   Self-supervised prompt optimization eliminates the need for labeled data by leveraging the intrinsic capabilities of LLMs to evaluate and refine prompts. Methods like PromptWizard (Agarwal et al., 2025) simulate labeled datasets by generating question-answer pairs, which are then used as pseudo-ground truth for optimization. However, the reliability of this approach heavily depends on the quality of the generated examples. Consistency-based methods, such as GLaPE (Zhang et al., 2024), evaluate prompts by clustering multiple outputs and scoring their consistency as Wang et al. (2023). This requires significant computational resources and often overlooks the reasoning process behind the outputs. Another approach, exemplified by SPO (Xiang et al., 2025), uses LLM-as-a-judge to compare responses of prompts based on user-defined criteria. Vague task specifications can limit the judgement reliability (Gu et al., 2025; Hu et al., 2024; Arabzadeh & Clarke, 2025). MR.PEA addresses these challenges by introducing a meta-reasoning-driven framework that dynamically builds task-specific knowledge, enabling robust and adaptive prompt optimization without relying on labeled data.

## 3 METHOD

### 3.1 PROBLEM SETUP AND PRELIMINARIES

MR.PEA is designed for prompt optimization in low-resource scenarios where labeled data is unavailable. It leverages meta-reasoning, the ability to reflect on and improve one's own reasoning process (Russell & Wefald, 1991; Yan et al., 2025), to iteratively build task-specific knowledge that guides optimization without ground-truth supervision.

**Minimal Input Requirements.**   Unlike supervised methods that require extensive labeled datasets, MR.PEA requires only three inputs: (1) Task Description $\tau$: a concise natural language description of the target task (e.g., "Infer the date from context."); (2) Initial Example $x_0$: a single unlabeled question of the task domain; (3) Format Specification $r$: instructions defining the desired output format (e.g., "Final answer in format `<answer>a multiple-choice option</answer>`.").

**Problem Formulation.**   Given the inputs $(\tau, x_0, r)$, MR.PEA aims to find the optimal prompt $p^*$ that maximizes task performance:

$$p^* = \arg\max_{p \in \mathcal{P}} \text{Performance}(p, \mathcal{X}), \tag{1}$$

where $\mathcal{P}$ represents the set of candidate prompts, and $\mathcal{X}$ denotes the set of validation examples generated during the optimization process.

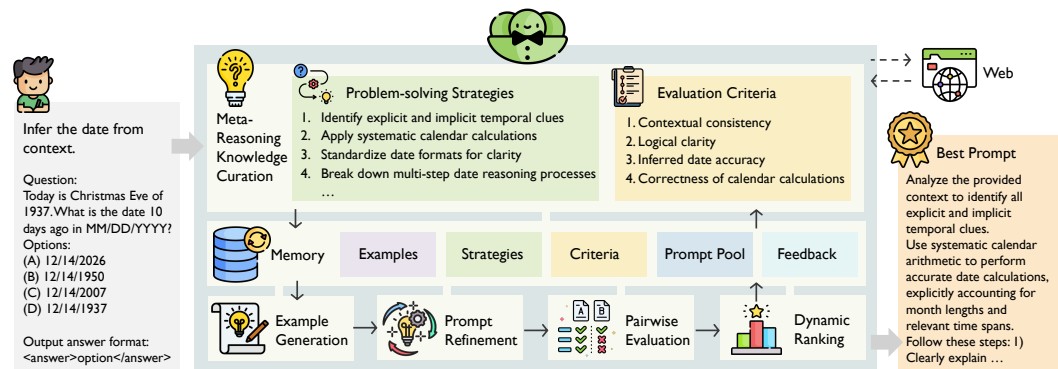

Figure 3: Overview of the MR.PEA framework (BBH *date_understanding* as an example). MR.PEA operates with a task description, a question, and a format specification. It iteratively curates task-specific knowledge, generates examples, refines prompts, evaluates their performance, and manages a dynamic prompt pool. The framework adaptively performs external web searches when needed. All components share a common memory to ensure knowledge consistency across phases.

**Meta-Reasoning Framework.** Meta-reasoning enables MR.PEA to dynamically construct task-specific knowledge $k_t$ at iteration $t$, comprising: (1) Abstract strategies: generalizable problem-solving techniques tailored to the task domain; (2) Evaluation criteria: reasoning-oriented assessment criteria that focus more on the process, emphasizing logical consistency, clarity, and task adherence. This part serves as the foundation for MR.PEA's optimization process to guide each phase of the refinement cycle.

## 3.2 META-REASONING-DRIVEN OPTIMIZATION CYCLE

MR.PEA employs a five-phase optimization cycle to iteratively refine prompts. During each iteration, it adaptively constructs and updates task-specific knowledge (phase 1), generates diverse validation examples (phase 2), refines candidate prompts (phase 3), evaluates them using reasoning-oriented criteria (phase 4), and updates the prompt ranking (phase 5). This process continues until convergence or the maximum iteration limit is reached. Figure 3 shows an overview of MR.PEA, and Algorithm 1 outlines the core procedures. Full algorithm is provided in Appendix 2. Below, we describe each phase of the optimization cycle in detail.

**Phase 1: Task-Specific Knowledge Curation.** The optimization process begins by constructing and refining task-specific knowledge to guide subsequent phases. At each iteration $t$, MR.PEA evaluates the current knowledge $k_{t-1}$ by analyzing the task description $\tau$ and the most recently generated example $x_{t-1}$. This evaluation identifies areas where the knowledge can be clarified or expanded (e.g., addressing ambiguities or adding missing insights). If no updates are required, the knowledge remains unchanged ($k_t = k_{t-1}$). When updates are needed, MR.PEA first refines the knowledge internally using its meta-reasoning capabilities. This involves deriving abstract strategies and reasoning-oriented evaluation criteria, which include generalizable problem-solving approaches (e.g., "break down complex problems into smaller, manageable steps" for math tasks or "use grammatical rules as primary guides" for language tasks) and corresponding process-focused evaluation criteria (e.g., "logical consistency of reasoning steps" or "consistency with grammatical rules").

To ensure the knowledge remains comprehensive, MR.PEA self-assesses the refined knowledge to evaluate whether it sufficiently addresses the task requirements and whether external information is required. This assessment examines knowledge completeness, task relevance, timing, impact potential, and resource efficiency. If gaps are identified, MR.PEA measures the importance of the missing knowledge and its confidence in the current decision. When both metrics exceed predefined thresholds, it initiates a web search, generates targeted queries, retrieves relevant insights, and integrates them into the existing knowledge. The number of external searches is constrained to $S_{\max}$ to ensure cost-effectiveness while benefiting from external information when needed. If external information is deemed unnecessary, it updates the knowledge solely through meta-reasoning.

---

**Algorithm 1** MR.PEA Optimization Process

---

**Require:** Task description $\tau$, initial unlabeled example $x_0$, format specification $r$
**Ensure:** Optimized prompt $p^*$
 1: Initialize initial prompt $p_0 \leftarrow \tau + r$, $p^* \leftarrow p_0$, knowledge $k_0 \leftarrow \emptyset$, feedback $f_0 \leftarrow \emptyset$,
 2: Initialize prompt pool with base ranking score $\mathcal{P} \leftarrow \{(p_0, \beta_{\text{base}})\}$, example memory $\mathcal{X} \leftarrow \{x_0\}$
 3: Set iteration count $t \leftarrow 1$, consecutive wins $w \leftarrow 0$
 4: **while** $t \leq T_{\max}$ and $w < T_{\text{win}}$ **do**
 5:     $k_t \leftarrow$ **MetaReason**$(\tau, k_{t-1}, x_{t-1})$
 6:     $x_t \leftarrow$ **GenerateExample**$(\tau, k_t, \mathcal{X})$                    ▷ Include latest examples
 7:     Update example memory $\mathcal{X} \leftarrow \mathcal{X} \cup \{x_t\}$
 8:     $p_t \leftarrow$ **RefinePrompt**$(\tau, k_t, x_t, f_{t-1}, \mathcal{P})$                ▷ Include top-ranking prompts
 9:     $(p_{\text{win}}, f_t) \leftarrow$ **PairwiseEvaluate**$(k_t, x_t, p^*, p_t)$
10:     $(p^*, w, \mathcal{P}) \leftarrow$ **ManagePromptPool**$(p_{\text{win}}, p^*, p_t, \mathcal{P})$        ▷ Rank prompts and update prompt pool
11: **end while**
12: **return** $p^*$

---

**Phase 2: Diverse Validation Example Generation.** To ensure robust internal evaluation and prevent overfitting, MR.PEA generates diverse validation examples including both questions and reference answers. The generation process systematically creates examples with structural and methodological diversity rather than superficial variations. It analyzes recently generated examples in memory $\mathcal{X}$ and dynamically varies new examples across multiple dimensions, including difficulty levels (ranging from basic to hard with extra twists), contextual variations (exploring different scenarios while maintaining task relevance), and structural diversity (testing alternative solution paths). This ensures that each new example $x_t$ contributes unique evaluation perspectives. The generated examples are added to the expanding example memory $\mathcal{X}$, forming a diverse validation pool. During evaluation, MR.PEA uses the latest examples to ensure prompt comparisons occur across varied questions, reducing the risk of overfitting to specific example patterns.

**Phase 3: Strategy-Guided Prompt Refinement.** The refinement process leverages accumulated task-specific knowledge $k_t$ and task context (i.e., task description $\tau$ and latest example $x_t$) to generate an improved prompt $p_t$. The refinement is guided by three complementary knowledge components: (1) Abstracted Strategies in $k_t$: task-specific problem-solving techniques that act as principled reasoning aids and are integrated to the refined prompt; (2) Historical Feedback $f_{t-1}$: insights from previous evaluations that highlight areas for improvement; (3) Top-Ranking Prompts in $\mathcal{P}$: successful templates that can be refined, combined, or adapted to uncover hidden improvement patterns. This approach ensures that each refinement builds on prior discoveries and targeted strategies, rather than relying solely on the LLM's free intuition. By incorporating task-specific insights, the process generates focused modifications, preserves successful elements, and systematically enhances prompt quality.

**Phase 4: Criteria-Based Pairwise Evaluation.** MR.PEA employs a structured evaluation process that combines pairwise prompt comparison with task-specific assessment criteria in $k_t$. It evaluates the newly refined prompt $p_t$ against the current best prompt $p^*$ using the latest validation example $x_t$. First, both prompts are executed on the validation question to produce responses $y^* = \phi(p^*, x_t)$ and $y_t = \phi(p_t, x_t)$, where $\phi$ represents execution with the underlying LLM. Then, MR.PEA assesses the quality of both responses using task-specific evaluation criteria from the knowledge base $k_t$, with the validation example $x_t$ serving as a reference. This ensures that the evaluation is grounded in the specific requirements of the task and the context provided by the latest example. It compares prompt-output pairs $(p^*, y^*)$ and $(p_t, y_t)$, determines the winner prompt $p_{win}$, and generates actionable feedback $f_t$ with specific, targeted, and constructive suggestions to guide the next iteration of prompt refinement: $p_{win}, f_t = \text{PairwiseEvaluate}((p^*, y^*), (p_t, y_t), k_t, x_t)$.

**Phase 5: Dynamic Prompt Pool Management.** The prompt pool is managed by a ranking system that employs adaptive scoring that rewards both immediate performance gain and sustained improvement. The update mechanism uses two strategies: (1) Exploration Incentive: when a new prompt $p_t$ wins, it receives a base score $\beta_{\text{base}}$ plus an iteration bonus that linearly grows with $t$: $\text{ranking\_score}(p_t) = \beta_{\text{base}} + t \times \alpha_{\text{bonus}}$. This encourages the active exploration of new prompts, prioritizing later-discovered prompts which often benefit from richer task-specific knowledge ac-

cumulated during earlier iterations. (2) Exploitation Reinforcement: when the current best prompt $p^*$ wins, all prompts in $\mathcal{P}$ have their scores reduced by a decay factor $\gamma$: ranking_score$(p_i) \leftarrow \gamma \cdot$ ranking_score$(p_i)$, $\quad \forall p_i \in \mathcal{P}$. Then the winning prompt $p^*$ receives an additional bonus (i.e., $t \times \alpha_{\text{bonus}}$) and the new prompt $p_t$ is assigned its base score. The decay prevents older prompts from dominating the ranking, and the bonus for $p^*$ encourages the retention of consistently high-performing prompts. The dual mechanism balances exploration of new prompts with exploitation of known strong performers, maintaining a competitive environment for new prompts to emerge.

After each evaluation, the prompt pool $\mathcal{P}$ is updated by adding the newly refined prompt to the pool and assigning new scores. The current best prompt $p^*$ is refreshed if necessary. The system continuously tracks consecutive wins to monitor convergence stability. The optimization terminates under two conditions: reaching the maximum iteration limit $T_{\max}$, or when the same prompt wins $T_{\text{win}}$ consecutive comparisons. This stopping criterion ensures efficient optimization without premature termination or excessive computational overhead.

## 4 EXPERIMENT

### 4.1 EXPERIMENTAL SETUP

**Benchmarks.** We use two benchmarks: (1) GSM8K (Cobbe et al., 2021): a dataset of grade-school multi-step arithmetic problems requiring 2–8 steps of reasoning, including addition, subtraction, multiplication, and division. (2) BIG-Bench Hard (BBH) (Suzgun et al., 2022): a challenging subset of the BIG-Bench (Srivastava et al., 2022) benchmark with 23 task types, covering a wide range of topics. We exclude *shuffled_objects* due to the content management policy of Azure Microsoft (2025b), and test on the remaining 22 types. We group BBH tasks into five sub-areas: Logic and Reasoning (*Logic*), Language and Semantics (*Language*), Mathematics and Arithmetic (*Math*), Common Sense and Factual Judgment (*Common Sense*), and Spatial, Sequential, and Attribute Reasoning (*Spatial/Seq./Attr.*). Details of the grouping are in the Appendix.

**Baselines.** We evaluate MR.PEA against three categories of baseline methods: (1) Static Prompts: Direct invocation and Zero-shot Chain-of-Thought (Kojima et al., 2022). (2) Supervised Prompt Optimization: OPRO (Yang et al., 2024) and GPO (Tang et al., 2025). (3) Self-Supervised Prompt Optimization: GLaPE (Zhang et al., 2024) (consistency-based), SPO (Xiang et al., 2025) (LLM-as-a-Judge), and PromptWizard (Agarwal et al., 2025) (synthetic supervision).

**Implementation Details.** We use GPT-4.1-nano (OpenAI, 2025) provided by Azure Microsoft (2025a) as the base LLM for prompt optimization with a maximum token limit of 1024, and use LinkUp (2025) for web search. We set the maximum iterations to 10, allow up to 2 web searches throughout the entire optimization process, use the 3 latest examples for generation, and retain the top 5 prompts for refinement. For ranking, we set the iteration bonus to 0.1 and decay factor to 0.9. Inputs strictly follow the minimal requirements described in Section 3.1: a concise task description sourced from the official dataset, a single unlabeled example (question only) randomly selected from the GSM8K training set or BBH CoT prompts, and a format specification "Final answer in `<answer>specific answer type</answer>`". Further details are in the Appendix.

**Evaluation Setup.** We use the same model as the optimization LLM described above. The evaluation is conducted in zero-shot settings with a temperature of 0.2 and a maximum token limit of 1500. The entire GSM8K test set and the full BBH dataset are used for testing. We re-implement all self-supervised baselines under our task settings and optimize using the same LLM to ensure fair comparisons. For static prompts and supervised prompt optimization methods, we also include the specified output format to maintain consistency across evaluations. Details are in the Appendix.

**Metrics.** We use accuracy as the evaluation metric, calculated as the percentage of answers that match the ground truth. To provide a comprehensive evaluation of model performance on answer correctness and format adherence, two parsing schemes are used: (1) *Strict Parsing:* the model's output must strictly follow the specified format. Only the final answers correctly enclosed within `<answer></answer>` tags and matching the ground truth are considered correct. (2) *Relaxed*

Table 1: Zero-shot accuracy on GSM8K and BBH sub-areas. Strict parsing results are shown in white rows, and relaxed parsing results are in shaded rows. The best results are highlighted in **bold**, the second best are underlined. MR.PEA's improvement over the best baseline is indicated in red.

| Method | GSM8K | Big-Bench Hard | | | | | Avg. |
|---|---|---|---|---|---|---|---|
| | | Common Sense | Language | Logic | Math | Spatial /Seq./Attr. | |
| **Static Prompts** | | | | | | | |
| Direct | 87.19 | 54.51 | 47.72 | 62.64 | 59.92 | 65.44 | 58.91 |
| | 91.74 | 66.99 | 58.73 | 77.52 | 68.56 | 79.16 | 70.51 |
| CoT | 85.29 | 51.17 | 60.57 | 69.68 | 68.16 | 74.35 | 65.22 |
| | 90.98 | 69.26 | 68.74 | 86.64 | 78.72 | 93.72 | 79.08 |
| **Supervised Prompt Optimization** | | | | | | | |
| OPRO | 90.98 | 62.06 | 53.08 | 65.20 | 63.28 | 60.33 | 65.02 |
| | 91.66 | 63.74 | 59.09 | 78.88 | 73.04 | 76.57 | 70.84 |
| GPO | 90.07 | 52.16 | 55.70 | 64.96 | 70.72 | 66.95 | 63.02 |
| | 90.07 | 59.60 | 64.04 | 77.12 | 75.68 | 88.58 | 73.06 |
| **Self-supervised Prompt Optimization** | | | | | | | |
| GLaPE | 81.50 | 54.64 | 50.40 | 59.04 | 59.92 | 62.53 | 58.06 |
| | 84.46 | 58.35 | 55.81 | 71.68 | 64.72 | 79.36 | 66.07 |
| SPO | 91.13 | 67.60 | 66.62 | 83.36 | 69.20 | 79.25 | 73.68 |
| | 91.74 | **72.38** | 70.44 | 88.48 | 78.48 | 90.59 | 80.09 |
| PW | 86.88 | 63.23 | 64.65 | 74.32 | 67.44 | 77.53 | 69.81 |
| | 87.26 | 63.58 | 67.33 | 86.56 | 83.36 | 86.13 | 77.42 |
| **MR.PEA (Ours)** | **92.34**+1.21 ↑1.3% | **69.87**+2.28 ↑3.4% | **67.55**+0.93 ↑1.4% | **85.84**+2.48 ↑3.0% | **84.00**+13.28 ↑18.7% | **87.23**+7.99 ↑10.1% | **79.10**+5.42 ↑7.4% |
| | **92.49**+0.75 ↑0.8% | 70.35-2.03 ↓2.8% | **72.39** +1.95 ↑2.8% | **89.60** +1.12 ↑1.3% | **87.60** +4.24 ↑5.1% | **95.39** +1.67 ↑1.8% | **82.94** +2.85 ↑3.6% |

*Parsing:* the model's output is considered correct if the final answer matches the ground truth, regardless of formatting. Unless otherwise specified, results reported are based on strict parsing.

## 4.2 MAIN RESULTS

Table 1 summarizes the zero-shot performance of MR.PEA compared to baselines across GSM8K and Big-Bench Hard (BBH) sub-areas, under both strict and relaxed parsing schemes.

MR.PEA consistently outperforms all baselines across all evaluated tasks, with an average improvement of 7.4% over the best baseline (SPO). It excels in tasks that require systematic reasoning and adherence to task-specific rules or procedures, demonstrating its ability to capture task-specific nuances. For example, our approach reaches 92.34% accuracy on GSM8K, surpassing the best baseline by 1.21%. In BBH *Math* tasks, it achieves 84.00% accuracy, a substantial 18.7% improvement. For *Spatial/Sequential/Attribute* reasoning, it attains 87.23% accuracy, a 10.1% gain over the best baseline. These results highlight MR.PEA's effectiveness in guiding models through complex, multi-step reasoning and strategic problem solving. MR.PEA also delivers consistent improvements in *Logic* (+3.0%) and *Language* (+1.4%) tasks, demonstrating broad applicability across diverse tasks.

Unlike previous self-supervised methods and other baselines that often compromise between answer quality and output format, MR.PEA achieves both high accuracy and strict format adherence. Under strict parsing, MR.PEA achieves an average accuracy of 79.10%, outperforming the best baseline by 7.4%. Even under relaxed parsing, it maintains its lead with 82.94% average accuracy, 3.6% higher than the best baseline (SPO). This demonstrates its ability to guide LLMs to produce answers that are both correct and consistently formatted, which is critical for downstream applications. Notably, MR.PEA achieves the highest accuracy in *Logic* (72.39%), *Math* (87.60%), and *Spatial/Sequential/Attribute* tasks (95.39%) under relaxed parsing. Although the performance gap narrows under relaxed parsing due to more permissive criteria, MR.PEA continues to lead in tasks that can be well-solved by applying generalizable rules.

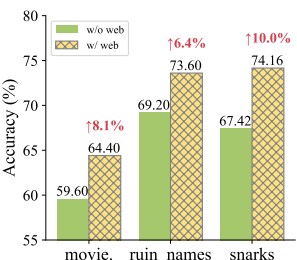

Figure 4: Cost-effectiveness of self-supervised methods on GSM8K.

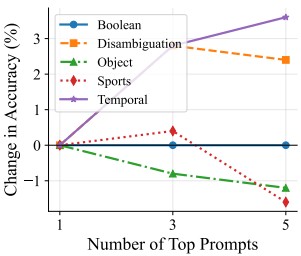

Figure 5: Impact of web search.

Figure 6: Impact of the number of historical prompts.

Table 2: Impact of MR.PEA components on task accuracy under strict parsing. The largest performance drops for each task are highlighted in **bold**, with red shading indicating the magnitude of the drop (darker shades represent larger drops). Tasks are categorized based on average output token length: easy★☆☆ (<200 tokens), moderate★★☆ (200–300 tokens), and hard★★★ (>300 tokens).

| **Removed Component** | boolean.★☆☆ | disambi.★★☆ | object.★☆☆ | sports.★★★ | temporal.★★★ |
|---|---|---|---|---|---|
| MR.PEA | 98.80 | 60.00 | 96.40 | 85.60 | 93.20 |
| w/o Abstract Strategies | 97.20 ↓1.6% | 55.20 ↓8.0% | 92.80 ↓3.7% | 70.00 ↓**18.2%** | 81.60 ↓**12.4%** |
| w/o Evaluation Criteria | 98.00 ↓0.8% | 57.20 ↓4.7% | 91.60 ↓**5.0%** | 78.40 ↓8.4% | 90.40 ↓3.0% |
| w/o Example Generation | 97.60 ↓1.2% | 59.60 ↓0.7% | 91.60 ↓**5.0%** | 70.80 ↓17.3% | 92.00 ↓1.3% |

MR.PEA is also highly cost-effective. As shown in Figure 4, MR.PEA's optimization is achieved at a cost of only $0.010 per task and completes each optimization in approximately 2 minutes. The cost is estimated according to Azure OpenAI pricing (Azure Microsoft, 2025c), based on the number of input and output tokens. More details are provided in the Appendix. While SPO is slightly faster (1.5 minutes) and comparable in cost ($0.011), it consistently underperforms MR.PEA, particularly in tasks requiring systematic reasoning such as *Math* and *Spatial/Sequential/Attribute* tasks. PromptWizard doubles the cost and requires more time per optimization, while GLaPE is the least practical, with a cost of $0.112 per task and an optimization time of 90 minutes, making it unsuitable for scalable deployment. These results underscore the advantage of MR.PEA, which delivers state-of-the-art performance while minimizing both financial and computational overhead.

## 4.3 ABLATION STUDIES

To understand the contribution of components within MR.PEA, we remove or modify key components of it and evaluate the impact. We select one representative task from each BBH category for ablation studies. Except for the web search ablation, all other ablations are performed without web search to isolate the effects of internal components.

**Effects of Meta-Reasoning Knowledge.** Meta-reasoning knowledge mainly guides the refinement and evaluation processes. We ablate abstract strategies for prompt refinement and evaluation criteria for quality assessment to study their individual contributions.

ABSTRACT STRATEGIES. Abstract strategies represent generalizable problem-solving techniques tailored to the task domain. They may include strategies like breaking down complex reasoning into manageable steps. Removing abstract strategies leads to significant performance drops, particularly in tasks requiring more reasoning, as shown in Table 2. For hard tasks like *sports_understanding* and *temporal_sequences*, accuracy drops by 18.2% and 12.4%. Even in moderate tasks like *disambiguation_qa*, the removal results in an 8.0% decrease. In contrast, easier tasks like *boolean_expressions* and *object_counting* see smaller declines (1.6% and 3.7%). This indicates that abstract strategies are more effective for complex problems but have limited impact on simpler ones.

EVALUATION CRITERIA. Evaluation criteria focus on assessing the quality of prompt and response pairs, including aspects like logical consistency, clarity, and adherence to task-specific rules. As shown in Table 2, removing evaluation criteria leads to moderate to significant performance degradation across tasks, particularly those requiring structured reasoning. For tasks like *sports_understanding*, the absence of evaluation criteria results in an accuracy drop of 8.4%. Similarly, in *disambiguation_qa*, accuracy decreases by 4.7%, and in *object_counting*, it decreases by 5.0%. In tasks with relatively straightforward structures, such as *boolean_expressions*, removing evaluation criteria only causes a 0.8% drop.

**Effects of Example Generation.** Example generation provides diverse and contextually relevant examples to test and refine prompts. Removing this component leads to varying degrees of performance degradation across tasks, as shown in Table 2. For tasks with a broader scope and more varied contexts, such as *sports_understanding*, the absence of example generation results in a significant accuracy drop of 17.3%. This highlights the importance of diverse examples in handling complex and nuanced reasoning. Similarly, in *object_counting*, accuracy decreases by 5.0%, indicating that example generation helps the model adapt to more possible scenarios. In contrast, tasks with simpler structures, such as *boolean_expressions*, show minimal performance degradation ($\downarrow$1.2%), suggesting that these tasks rely less on diverse examples and are less sensitive to the removal of this component. For *disambiguation_qa*, the drop is relatively small ($\downarrow$0.7%), indicating that while examples are helpful, this task depends more on evaluation criteria and strategic refinement.

**Effects of Web Search.** In most cases, MR.PEA can optimize prompts effectively without any external information. It relies on its internal knowledge and meta-reasoning abilities. However, for tasks that it cannot resolve by itself, seeking external knowledge via web search can provide crucial context that enhances its understanding. We ablate the only three tasks that benefit from web search.

Figure 5 shows the impact of web search on MR.PEA's performance for BBH tasks requiring external knowledge. Web search significantly improves accuracy, particularly for tasks needing factual information. In the *movie_recommendation* task, accuracy increases from 59.6% to 64.4% (8.1% improvement). In the *ruin_names* task, it rises from 69.2% to 73.6% (6.4% improvement). Similarly, in the *snarks* task, accuracy improves from 67.42% to 74.16% (10.0% improvement). These results demonstrate the value of external knowledge in enhancing model performance. Specifically, in the *ruin_names* task, MR.PEA searches for "methods for generating humorous one-character word edits with valid words". It enhances its knowledge with externally retrieved techniques such as "constructing a distance graph to explore neighbors of a given word" (i.e., words differing by one letter) and "combining funny, weird, or silly words to select humorous yet valid substitutions".

**Effects of the Number of Historical Prompts Retained.** The ranking system in MR.PEA enables the selection of top prompts as references during refinement. Figure 6 shows how the number of top-ranking historical prompts impacts task accuracy. Increasing the number of historical prompts from 1 to 3 leads to noticeable improvements across most tasks, such as a 2.8% gain in *disambiguation_qa* and *temporal_sequences*. However, increasing the number to 5 yields only marginal improvements or slight fluctuations. Complex tasks like *disambiguation_qa* and *temporal_sequences* benefit more from additional historical prompts due to the richer context and patterns they provide. In contrast, simpler tasks like *boolean_expressions* show little to no improvement. These findings suggest maintaining a moderate number of historical prompts allows the model to effectively balance leveraging useful context and avoiding excessive noise.

## 5 CONCLUSION

We introduced MR.PEA, a self-supervised framework for prompt optimization that leverages meta-reasoning to tackle the challenges of low-resource scenarios. By iteratively constructing task-specific knowledge, including problem-solving strategies and reasoning-oriented evaluation criteria, MR.PEA eliminates optimization ambiguity and enhances prompt refinement without the need for labeled data. Its ability to adaptively curate knowledge, generate diverse validation examples, and manage a dynamic prompt pool ensures robust and scalable optimization. Experimental results demonstrate that MR.PEA is both highly effective and efficient in terms of cost and computation.

## REPRODUCIBILITY STATEMENT

To ensure reproducibility, we evaluate our approach on publicly available benchmarks. All experiments are performed using the Azure OpenAI API with GPT-4.1-nano (model version 2025-04-14). We include detailed algorithm designs in pseudo code, the prompts used in MR.PEA, all input data, experimental settings, and the optimized prompts in the appendix.

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

# A  APPENDIX

## LLM USAGE STATEMENT

We used GPT-4o to assist in the preparation of this paper. We used it to check grammar, improve sentence fluency and generate table templates.

# B  EXPERIMENT SETTINGS

## B.1  BIG-BENCH HARD GROUPING

We divided the BIG-Bench Hard (BBH) tasks into five major categories to balance the number of tasks in each group and to reflect the main types of reasoning or skills required. The categories are:

• **Common Sense and Factual Judgment:** Tasks that test common sense, factual knowledge, and the ability to detect errors or lies, including *causal_judgement, movie_recommendation, salient_translation_error_detection, sports_understanding, web_of_lies.*

• **Language and Semantics:** Tasks that require understanding language structure, semantics, and word manipulation, including *disambiguation_qa, hyperbaton, ruin_names, snarks, word_sorting.*

• **Logic and Reasoning:** Tasks focused on logical deduction, formal reasoning, and evaluating logical expressions, including *boolean_expressions, formal_fallacies, logical_deduction_five_objects, logical_deduction_seven_objects, logical_deduction_three_objects.*

• **Mathematics and Arithmetic:** Tasks involving mathematical calculations, counting, and geometric reasoning, including *date_understanding, dyck_languages, geometric_shapes, multi-step_arithmetic_two, object_counting.*

• **Spatial, Sequential, and Attribute Reasoning:** Tasks related to spatial relationships, sequences, and reasoning about object attributes, including *navigate, penguins_in_a_table, reasoning_about_colored_objects, temporal_sequences.*

## B.2  MINIMAL INPUT EXAMPLE

Below we show all the inputs used in our experiments. Each input consists of three parts: task description, format requirement, and a sample question. For BBH, we show multiple examples since different tasks have different answer formats.

Listing 1: GSM8k input example

```
task_description: "Solve grade-school math word problems that require multi-step reasoning and
    the application of basic arithmetic operations (addition, subtraction, multiplication,
    and division). May include intermediate calculations to arrive at the final numeric
    answer."
format_requirement: "Final answer in format <answer>a number</answer>."
sample_question: |
    Jenny likes to play board games with her friends.  She likes to play against her friend
        Mark the most, as she's played him 10 times and Mark has only won once.  She doesn't
        like playing Jill, as she's played her twice as many times as she's played Mark and
        Jill has won 75% of them. How many times in total has Jenny won board games with her
        two friends?
```

Listing 2: BBH input example - true or false

```
boolean_expressions:
    task_description: "Evaluate the result of a random Boolean expression."
    format_requirement: "Final answer in format <answer>true or false</answer>."
    sample_question: |
    True and False and not True and True is
```

Listing 3: BBH input example - yes or no

```
sports_understanding:
    task_description: "Determine whether an artificially constructed sentence relating to
        sports is plausible or not."
```

```
format_requirement: "Final answer in format <answer>yes or no</answer>."
sample_question: |
Is the following sentence plausible? "Bam Adebayo scored a reverse layup in the Western
    Conference Finals."
```

Listing 4: BBH input example - multiple choice

```
date_understanding:
    task_description: "Infer the date from context."
    format_requirement: "Final answer in format <answer>a multiple-choice option</answer>."
    sample_question: |
    Today is Christmas Eve of 1937. What is the date 10 days ago in MM/DD/YYYY?
    Options:
    (A) 12/14/2026
    (B) 12/14/1950
    (C) 12/14/2007
    (D) 12/14/1937
    (E) 07/14/1938
    (F) 12/14/1988
```

Listing 5: BBH input example - string

```
dyck_languages:
    task_description: "Correctly close a Dyck-n word."
    format_requirement: "Final answer in format <answer>your answer</answer>."
    sample_question: |
    Complete the rest of the sequence, making sure that the parentheses are closed properly.
        Input: < > ( ( [ [ ( ( { } ) [ < > ] ]
```

Listing 6: BBH input example - valid or invalid

```
formal_fallacies:
    task_description: "Distinguish deductively valid arguments from formal fallacies."
    format_requirement: "Final answer in format <answer>valid or invalid</answer>."
    sample_question: |
    "It is not always easy to see who is related to whom -- and in which ways. The following
        argument pertains to this question: To begin with, Lesley is a close friend of
        Fernando. Moreover, being a close friend of Fernando or a schoolmate of Lowell is
        sufficient for being a great-grandfather of Leroy. It follows that Lesley is a great-
        grandfather of Leroy."
    Is the argument, given the explicitly stated premises, deductively valid or invalid?
    Options:
    - valid
    - invalid
```

Listing 7: BBH input example - number

```
object_counting:
    task_description: "Questions that involve enumerating objects and asking the model to
        count them."
    format_requirement: "Final answer in format <answer>a number</answer>."
    sample_question: |
    I have an orange, a raspberry, two peaches, a blackberry, an apple, a grape, a nectarine,
        and three plums. How many fruits do I have?
```

Listing 8: BBH input example - sorted list

```
word_sorting:
    task_description: "Sort a list of words."
    format_requirement: "Final answer in format <answer>sorted word list</answer>."
    sample_question: |
    Sort the following words alphabetically: List: hypochlorite ponderosa phone credulity
```

### B.3 SELF-SUPERVISED BASELINES

For self-supervised baselines, we use similar inputs as our method, with slight modifications to adapt to their respective settings. Below are their detailed configurations, using *boolean_expressions* from BBH as an example:

**Ours.** We use the minimal input as shown below. During evaluation, our optimized prompts are used as the system prompt, and questions are used as the user prompt.

Listing 9: MR.PEA input example

```
boolean_expressions:
    task_description: "Evaluate the result of a random Boolean expression."
    format_requirement: "Final answer in format <answer>true or false</answer>."
    sample_question: |
    True and False and not True and True is
```

**GlaPE.** GlaPE is to optimize a CoT trigger. It starts with a base instruction "Let's think step by step,". The original implementation also includes an answer trigger to extract the answer from the generated reasoning. The input format is `input = "Q: " + question + "\nA: " + instruction`. We keep format consistent in evaluation. For GSM8K, the optimization iteration is set to 10, the number of candidates generated per iteration is 8, and the number of answers for self-consistency check is set to 10, following the default settings provided by the authors. For BBH, the optimization iteration is set to 5, the number of candidates generated per iteration is 3, and the number of answers for self-consistency check is set to 5. The input is shown below.

Listing 10: GlaPE input example

```
base_instruction: "Let's think step by step."

task:
    boolean_expressions:
        task_description: "Evaluate the result of a random Boolean expression."
        format_requirement: "Final answer in format <answer>true or false</answer>."
        sample_question: |
        Q: True and False and not True and True is
        answer_trigger: |
        Therefore, the answer (true or false) is
```

**SPO.** Following the original implementation of SPO, we futher include "The provided prompt needs to adapt to all current types of questions." and "A lot of thinking and analysis processes." in input. The optimization process is configured to run for 10 iterations for all datasets, following the default settings provided by the authors. We use their optimize prompt as the system prompt during evaluation and questions are user prompt. The input is shown below.

Listing 11: SPO input example

```
prompt: |
  Evaluate the result of a random Boolean expression.
  Final answer in format <answer>true or false</answer>.

requirements: |
  Final answer in format <answer>true or false</answer>.
  The provided prompt needs to adapt to all current types of questions.

count: None

qa:
  - question: |
        True and False and not True and True is

    answer: |
          A lot of thinking and analysis processes.
          ...
          Final Answer:
          <answer>true or false</answer>
```

**PromptWizard.** We set the synthetic examples to be 10, similar to our method. In PromptWizard, the sample question is used only for generating synthetic examples. We set the initial prompt to "Let's think step by step", as suggested by the authors. PromptWizard uses `<ANS_START>` and `<ANS_END>` to identify the answer part, and we adjust the format requirement accordingly. For GSM8K, the settings include 3 style variations, 3 mutation-refinement iterations, 3 mutation rounds, and 3 task example refinement iterations, following their default settings. For BBH, these settings except for style variation are reduced to 2. PromptWizard optimizes both role instruction and task

instruction. During experiment, we use their optimize role instruction as the system prompt and task instruction plus questions as the user prompt. The input is shown below.

Listing 12: PromptWizard input example

```
base_instruction: "Let's think step by step."

task:
    boolean_expressions:
        task_description: "Evaluate the result of a random Boolean expression."
        format_requirement: "Final answer in format <ANS_START>true or false<ANS_END>."
        sample_question: |
            True and False and not True and True is
```

### B.4 SUPERVISED BASELINES

For supervised methods OPRO and GPO, we utilize the optimized prompts provided by their respective authors. We adopt their default settings, where the input instruction follows the question in the format: `input = question + "\n" + instruction`. Both methods use 20% of the BBH dataset as the validation set for prompt optimization. Specifically, OPRO employs the GPT-3.5-turbo optimizer for prompt optimization, and GPO leverages the GPT-4 optimizer.

## C  CASE STUDY

We present two case studies to illustrate the capabilities of MR.PEA in different scenarios. The first case shows situations where MR.PEA relies solely on its internal knowledge even without requiring frequent updates during iterations. The second case shows scenarios where MR.PEA leverages external knowledge through web searches to address tasks with insufficient internal knowledge. We exclude the format requirements from the prompts in both cases for brevity.

### C.1  CASE 1: KNOWLEDGE SUFFICIENT - GSM8K

In the GSM8K task, MR.PEA updates its knowledge only once at the beginning and continues to use the same knowledge throughout all subsequent iterations. This knowledge includes specific reasoning patterns designed to solve the problems, as detailed below. Although these reasoning patterns are not explicitly provided in the input, MR.PEA effectively identifies and applies relevant strategies. The evaluation criteria focus not only on the correctness of the final answer but also on the logical consistency of the reasoning steps, the accuracy of intermediate calculations, and the clarity and completeness of the problem-solving process.

---

**GSM8K Knowledge**

Problem-solving strategies:
- "Break down complex problems into smaller, manageable steps";
- "Identify key information and relevant data points";
- "Use intermediate calculations to simplify multi-step reasoning";
- "Apply basic arithmetic operations systematically";
- "Establish a clear plan before performing calculations";
- "Verify each step for accuracy before proceeding";

Evaluation criteria:
- "Correctness of the final answer";
- "Logical consistency of reasoning steps";
- "Accuracy of intermediate calculations";
- "Clarity and completeness of the problem-solving process";
- "Ability to generalize approach to different problems";
- "Efficient use of arithmetic operations";

---

The best prompt generated by MR.PEA in the 7th iteration and the second-best prompt from the 4th iteration are presented below. We can see both prompts incorporate the knowledge provided above. The second best prompt is longer, and the best prompt is more concise and to the point.

---

**GSM8K Best Prompt (7th iteration)**

Analyze the given grade-school math word problem by identifying key data points. Break it into clear, sequential steps, performing all intermediate calculations explicitly with proper notation. Verify each step for accuracy. Use consistent terminology and formatting. Summarize the reasoning process briefly, explicitly connect each step to the question, and conclude with the final answer, ensuring clarity, correctness, and completeness.

---

**GSM8K Second Best Prompt (4th iteration)**

Analyze the given grade-school math word problem by identifying key data points. Break down the problem into clear, sequential steps, performing necessary intermediate calculations with consistent formatting. Include brief explanations of each step's significance to enhance understanding. Verify each intermediate result for accuracy before proceeding. Use variables and logical flow to ensure clarity and completeness. Conclude with the final answer, explicitly addressing the question's context and any assumptions or implications.

---

## C.2 CASE 2: KNOWLEDGE INSUFFICIENT - BBH RUIN_NAME

In the BBH ruin_name task, MR.PEA conducts web searches at iteration 2 and 3 to supplement its internal knowledge. At iteration 2 it searches "examples of humorous one-character edits creating valid words", but it does not find any good examples. The search at iteration 3 query "methods for generating humorous one-character word edits with valid words", returns several methods that can be incorporated into the knowledge base.

---

**BBH ruin_names Knowledge from Web Search**

- Construct a graph where nodes are valid words and edges connect words differing by exactly one character (edit distance of 1). By exploring neighbors of a given word (words differing by one letter), you can find valid one-character edits that are real words. This method ensures all intermediate words are valid.
- For each character position in the original word, substitute it with every possible letter of the alphabet and check against a dictionary or word list to verify if the resulting word is valid. This can be done efficiently using a hash table or preprocessed dictionary for O(1) lookups after O(n) preprocessing.
- Combine the above with generators of funny, weird, or silly words to select humorous valid words for substitution. Tools like random funny word generators or weird word generators can provide amusing candidates.
- Combine the above with generators of funny, weird, or silly words to select humorous valid words for substitution. Tools like random funny word generators or weird word generators can provide amusing candidates.

---

Then MR.PEA integrates this new knowledge to the problem-solving strategies. Below shows the knowledge before and after the web search. We can see that the problem-solving strategies have been enriched with new methods, and the evaluation criteria now include an additional focus on humor and double meanings.

> **BBH ruin_names Knowledge Before Search**
>
> Problem-solving strategies:
> - "Identify key words in the name and consider common letter substitutions or insertions."
> - "Leverage humor by replacing or adding letters to create puns or double meanings."
> - "Ensure the edited name remains an actual word after modification."
> - "Use phonetic similarities to generate humorous variants."
> - "Apply a step-by-step process: select critical letters, test modifications, evaluate humor and word validity."
> Evaluation criteria:
> - "The edited name is a valid word."
> - "The name change alters or humorizes the original meaning."
> - "The modified name retains recognizability."
> - "The edit involves only one character."
> - "The change produces a humorous or pun-like effect."

> **BBH ruin_names Knowledge After Search**
>
> Problem-solving strategies:
> - "Build a graph of valid words with edges representing single-character differences and explore neighbors."
> - "Substitute each character with alphabet letters and verify validity via a dictionary for quick filtering."
> - "Use word generators and pattern-matching tools to find close, humorous valid words."
> - "Combine word substitution with humorous word lists or generators for amusing options."
> - "Apply recursive backtracking or shortest path algorithms to find sequences of humorous, valid one-character edits."
> Evaluation criteria (including previous ones with add-on):
> - "The edit introduces humor, puns, or double meanings."

Comparing the best prompt at the 8th iteration with the prompt before web search at the 1st iteration, we can see that the best prompt is more detailed, incorporating "dictionary lookups" found during the search. It emphasizes more on clarity, validity, and humor effectiveness.

> **BBH ruin_names Best Prompt (8th iteration)**
>
> Given an artist, band, or movie name, identify a single-character substitution, insertion, or deletion that results in a valid, dictionary-recognized word. The new word should humorously alter or play on the original name, creating a pun, double meaning, or humorous twist, while remaining recognizable. Use dictionary lookup to verify the validity of the new word. Clearly specify the original name, the type of edit (substitution, insertion, or deletion), and the exact character change. Describe briefly how the modification introduces humor and why the new word maintains core recognition of the original. Ensure the change produces a real, humorous, and related word, focusing on clarity, validity, and humor effectiveness.

> **BBH ruin_names Prompt before Web Search (1st iteration)**
>
> Given an artist, band, or movie name, identify a single-character change—either substitution, insertion, or deletion—that transforms the name into a different, valid word. The modified word should humorously alter or play on the original meaning, creating a pun or double entendre. Ensure the change maintains recognizability of the original name, results in a legitimate word, and produces a humorous effect. Clearly specify the original name, the type of change (substitution, insertion, or deletion), and the resulting word. Do not provide examples; focus on clear, step-by-step reasoning to find the best modification.

We demonstrate a simple case on how web search can help improve the knowledge curation. While web search is a valuable tool for supplementing internal knowledge, its effectiveness depends on crafting precise and relevant queries. For uncommon tasks, it can be difficult to create queries that yield useful results. Additionally, the vast amount of information on the web makes it challenging

to locate the right resources and assess the quality of the retrieved knowledge. The wrong information may even mislead the model and lead to worse performance. Better knowledge augmentation methods can be explored to enhance the prompt optimization process.

### C.3 Using Generated Examples as Few-Shot Examples

We conducted experiments to evaluate the impact of using generated examples and answers as few-shot examples. The evaluation was performed on a GSM8K subset consisting of 100 randomly-selected samples. We selected examples generated in the last three iterations as few-shot examples. As the generated examples do not address the final answer format, we further include the final answer in the targeted answer format. In the experiment, we used the best prompt generated by MR.PEA in the 7th iteration (as shown above) and these generated examples to form a few-shot setting.

The results show that incorporating few-shot examples improves performance. Using only the best prompt achieves an accuracy of 91/100. When the three generated examples are added, the accuracy increases to 96/100. This in term demonstrates the quality of the examples generated by MR.PEA.

Below are the three examples generated by MR.PEA that were used as few-shot examples. In addition to changing the numbers, the examples cover different contexts or involve varying arithmetic steps while all focusing on the percentage problems. Specifically, from iteration 8 to 9, it changed context from flowers to books; from iteration 9 to 10, it changed context from books to children's books and added an additional calculation step.

---

**Examples Generated by MR.PEA**

Iteration 8:
Question: A gardener plants 180 flower bulbs. Of these, 2/3 are tulips, and the rest are daffodils. If 80% of the tulips bloom successfully, how many tulips bloom? If next season, the gardener plants 25% more tulips than this season, how many tulips will be planted then?
Answer: Total bulbs: 180. Tulips: (2/3)*180=120. Blooming tulips: 80% of 120=0.80*120=96. Next season, tulips planted: 25% more than 120=120 + (0.25*120)=150.

Iteration 9:
Question: A school organizes a book donation drive collecting 400 books. If 60% of these are fiction books, and 25% of the fiction books are damaged and need replacement, how many fiction books need replacement? If the school plans to increase fiction book donations by 20% next year, how many fiction books will they collect then?
Answer: First, find the number of fiction books: 60% of 400=0.60*400=240. Damaged fiction books: 25% of 240=0.25*240=60. Next year, fiction donations increase by 20%: 240 + (0.20*240)=240 + 48=288 books.

Iteration 10:
Question: A bookstore sells 1200 books each month. Of these, 1/4 are children's books. If 10% of the children's books are returned unsold at the end of the month, how many children's books are returned? If the bookstore plans to increase total sales by 15% next month, assuming the proportion of children's books remains the same, how many children's books will they sell next month?
Answer: Total books sold: 1200. Children's books: (1/4)*1200=300. Returned books: 10% of 300=0.10*300=30. Next month sales: 15% increase: 1200 + (0.15*1200)=1380. Children's books next month: 1380*(1/4)=345.

---

## D  Full Algorithm of MR.PEA

### D.1  MR.PEA: Core Algorithm Pseudocode

We present the complete pseudo code of MR.PEA below.

**Algorithm 2** Meta-Reasoning Prompt Engineering Agent (MR.PEA) Full Algorithm

**Require:** Task description $\tau$, format requirements $r$, initial example $x_0$
**Ensure:** Optimized prompt $p^*$
1: **Initialize:**
2: Initialize prompt $p_0 \leftarrow \tau + r$, best prompt $p^* \leftarrow p_0$, knowledge $k_0 \leftarrow \emptyset$, feedback $f_0 \leftarrow \emptyset$
3: Initialize prompt pool $\mathcal{P} \leftarrow \{(p_0, \beta_{\text{base}})\}$ with base ranking scores
4: $M \leftarrow$ INITIALIZEMEMORY$(k_0, x_0, f_0, \mathcal{P})$   ▷ Knowledge, examples, feedback, prompt pool
5: Set parameters: importance $\theta_{\text{imp}}$, confidence $\theta_{\text{conf}}$, bonus factor $\alpha_{\text{bonus}}$, decay factor $\gamma$
6: Set counters: $t \leftarrow 1$, consecutive wins $w \leftarrow 0$, search count $s \leftarrow 0$
7: **Main Optimization Loop:**
8: **while** $t \leq T_{\text{max}}$ **and** $w < T_{\text{win}}$ **do**
9:
10:    **Phase 1: Task-Specific Knowledge Curation**
11:    $k_{\text{temp}} \leftarrow$ METAREASON$(\tau, M.\text{knowledge}, M.\text{example})$
12:    **if** $k_{\text{temp}} = M.\text{knowledge}$ **or** $k_{\text{temp}}.\text{needs\_update} =$ FALSE **then**
13:     $k_t \leftarrow M.\text{knowledge}$          ▷ No knowledge update needed
14:    **else**
15:     **if** $t \geq 2$ **and** $s < S_{\text{max}}$ **then**    ▷ Conditional web search from 2nd iteration
16:      decision $\leftarrow$ METADECISION$(\tau, k_{\text{temp}}, t, s)$
17:      $I \leftarrow$ decision.importance, $C \leftarrow$ decision.confidence     ▷ Scores
18:      **if** decision.search\_needed **and** $I > \theta_{\text{imp}}$ **and** $C > \theta_{\text{conf}}$ **then**
19:       $q \leftarrow$ GENERATEQUERY$(\tau, \text{decision.gap})$
20:       $k_{\text{web}} \leftarrow$ WEBSEARCH$(q)$
21:       $k_{\text{temp}} \leftarrow$ INTEGRATEKNOWLEDGE$(k_{\text{temp}}, k_{\text{web}}), s \leftarrow s + 1$
22:      **end if**
23:     **end if**
24:     $k_t \leftarrow k_{\text{temp}}, M.\text{knowledge} \leftarrow M.\text{knowledge} \cup \{k_t\}$
25:    **end if**
26:
27:    **Phase 2: Diverse Validation Example Generation**
28:    $\mathcal{X} \leftarrow$ GETRECENTEXAMPLES$(M.\text{examples}, n)$     ▷ Include n latest examples
29:    $x_t \leftarrow$ GENERATEEXAMPLE$(\tau, k_t, \mathcal{X})$
30:    $M.\text{examples} \leftarrow M.\text{examples} \cup \{x_t\}$
31:
32:    **Phase 3: Strategy-Guided Prompt Refinement**
33:    $\mathcal{H} \leftarrow$ GETTOPPROMPTS$(\mathcal{P}, m)$       ▷ Include m historical top prompts
34:    $p_t \leftarrow$ REFINEPROMPT$(\tau, k_t, x_t, M.\text{feedback}, \mathcal{H})$
35:
36:    **Phase 4: Criteria-Based Pairwise Evaluation**
37:    $y^* \leftarrow \phi(p^*, x_t), y_t \leftarrow \phi(p_t, x_t)$       ▷ Execute prompts
38:    $(p_{\text{win}}, f_t) \leftarrow$ PAIRWISEEVALUATE$((p^*, y^*), (p_t, y_t), k_t, x_t)$
39:    $M.\text{feedback} \leftarrow M.\text{feedback} \cup \{f_t\}$
40:
41:    **Phase 5: Dynamic Prompt Pool Management**
42:    **if** $p_{\text{win}} = p_t$ **then**            ▷ New prompt wins
43:     score$(p_t) \leftarrow \beta_{\text{base}} + t \cdot \alpha_{\text{bonus}}$
44:     $p^* \leftarrow p_t, w \leftarrow 0$          ▷ Reset consecutive wins
45:    **else**                ▷ Current best wins
46:     $\forall p \in \mathcal{P} :$ score$(p) \leftarrow \gamma \cdot$ score$(p)$       ▷ Apply decay
47:     score$(p^*) \leftarrow$ score$(p^*) + t \cdot \alpha_{\text{bonus}}$    ▷ Boost the current best prompt
48:     score$(p_t) \leftarrow \beta_{\text{base}}, w \leftarrow w + 1$    ▷ Baseline score for new prompt
49:    **end if**
50:    $\mathcal{P} \leftarrow \mathcal{P} \cup \{(p_t, \text{score}(p_t))\}, t \leftarrow t + 1$      ▷ Add to prompt pool
51: **end while**
52: **return** $\arg\max_{p \in \mathcal{P}}$ score$(p)$

## D.2 KEY ALGORITHMIC COMPONENTS

**Multi-Reasoning Modules.** MR.PEA consists of six reasoning modules that work collaboratively:

$$
\begin{aligned}
\text{Modules} = \{ &\text{METAREASON} : (\tau, k_{t-1}, x_{t-1}) \to k_t, \\
&\text{METADECISION} : (\tau, k_t, t, s) \to (\text{search\_needed}, I, C), \\
&\text{GENERATEQUERY \& WEBSEARCH} : (\tau, \text{gap}) \to q, q \to k_{\text{web}}, \\
&\text{GENERATEEXAMPLE} : (\tau, k_t, \mathcal{X}) \to x_t, \\
&\text{REFINEPROMPT} : (\tau, k_t, x_t, f_{t-1}, \mathcal{H}) \to p_t, \\
&\text{PAIRWISEEVALUATE} : ((p^*, y^*), (p_t, y_t), k_t, x_t) \to (p_{\text{win}}, f_t), \}
\end{aligned}
$$

**Memory Architecture.** The memory system $M$ maintains persistent state across iterations and supports efficient retrieval operations: $M.\text{knowledge}$ returns the latest consolidated knowledge, $M.\text{example}$ provides the most recent example, and $\mathcal{P}$ maintains dynamic rankings.

$$
\begin{aligned}
M = \{ &\text{knowledge} : \bigcup_{i=0}^{t} \{k_i\} \text{ (task-specific strategies and evaluation criteria)}, \\
&\text{examples} : \{x_i\}_{i=0}^{t} \text{ (generated validation examples)}, \\
&\text{feedback} : \{f_i\}_{i=1}^{t-1} \text{ (evaluation insights)}, \\
&\text{prompts} : \mathcal{P} = \{(p_i, \text{score}(p_i))\}_{i=0}^{t} \text{ (ranked prompt pool)}, \}
\end{aligned}
$$

# E FULL EXPERIMENT RESULTS

## E.1 COST COMPARISON

We provide a detailed cost comparison of self-supervised methods on GSM8K in Table 3. All optimizations follow the authors' default settings, with inputs adapted to match our method. The comparison includes the number of API calls, input tokens, output tokens, estimated price cost, and optimization time. All methods utilize GPT-4.1-nano as the base LLM, and the price cost is estimated based on Azure OpenAI's pricing ($0.10 per 1M input tokens and $0.40 per 1M output tokens). Our method strikes a good balance between cost and performance, incurring only slightly higher costs than SPO while delivering significantly better performance. We draw Figure 4 in the main paper based on this table.

Table 3: Cost comparison of self-supervised methods on GSM8K.

| Method | API Calls | Input Tokens | Output Tokens | Price Cost ($USD) | Time (minutes) |
|---|---|---|---|---|---|
| GLaPE | 1660 | 322,747 | 198,385 | 0.112 | ∼90 |
| SPO | **55** | 67,946 | **10,744** | 0.011 | **∼1.5** |
| PromptWizard | 154 | **39,385** | 46,671 | 0.023 | ∼12 |
| MR.PEA (Ours) | 57 | 48,519 | 12,173 | **0.010** | ∼2 |

## E.2 FULL BBH RESULTS

We present the complete results for each BBH task in Table 4. These results form the basis for the calculations in Table 1, where we summarize the average accuracy across the task groups. For Dyck languages, since GPT-4.1-nano often generates the entire sequence rather than just the missing portion to be completed, we evaluate its output by checking whether the generated sequence includes the ground truth sequence as a substring. Other tasks are evaluated using exact match.

We also present the number of successfully parsed cases under strict evaluation in Table 5. MR.PEA achieves the highest average strict parsing rate of 95.25%, surpassing all other baselines. This also corresponds to MR.PEA's superior performance in strict parsing evaluation.

Table 4: Complete BBH results on each task. The strict parsing results are shown in white rows, and the loose parsing results are in light gray rows with improvements annotated. * means web search is used during optimization. Others are using self-knowledge only.

| Task | Direct | CoT | OPRO | GPO | GLaPE | SPO | PW | MR.PEA |
|---|---|---|---|---|---|---|---|---|
| **Common Sense and Factual Judgment** | | | | | | | | |
| causal_judgement | 59.36 | 50.27 | 55.08 | 58.82 | 57.22 | 45.99 | 56.15 | **62.57** |
| | 59.36 | 61.50 | 55.08 | 58.82 | 57.75 | 56.68 | 56.68 | 62.57 |
| | +0.00 | +11.23 | +0.00 | +0.00 | +0.53 | +10.69 | +0.53 | +0.00 |
| movie_recommendation | 37.60 | 37.20 | 67.60 | 46.00 | 66.40 | **68.40** | 57.20 | 64.40* |
| | 66.40 | 62.40 | 68.00 | 59.60 | 71.20 | 78.80 | 57.20 | 64.80 |
| | +28.80 | +25.20 | +0.40 | +13.60 | +4.80 | +10.40 | +0.00 | +0.40 |
| salient_translation_error_detection | 44.00 | 47.60 | 41.60 | 32.40 | 40.80 | **55.20** | 40.00 | 52.00 |
| | 48.40 | 56.40 | 48.80 | 53.60 | 50.80 | 57.60 | 40.80 | 53.20 |
| | +4.40 | +8.80 | +7.20 | +21.20 | +10.00 | +2.40 | +0.80 | +1.20 |
| sports_understanding | 64.40 | 44.40 | 65.20 | 65.60 | 63.60 | 72.00 | **88.00** | 84.00 |
| | 64.40 | 69.20 | 65.20 | 65.60 | 66.80 | 72.40 | 88.40 | 84.80 |
| | +0.00 | +24.80 | +0.00 | +0.00 | +3.20 | +0.40 | +0.40 | +0.80 |
| web_of_lies | 67.20 | 76.40 | 80.80 | 58.00 | 45.20 | **96.40** | 74.00 | 86.40 |
| | 96.40 | 96.80 | 81.60 | 60.40 | 45.20 | 96.40 | 74.40 | 86.40 |
| | +29.20 | +20.40 | +0.80 | +2.40 | +0.00 | +0.00 | +0.04 | +0.00 |
| **Language and Semantics** | | | | | | | | |
| disambiguation_qa | 52.00 | 37.20 | 54.80 | 45.60 | 40.40 | 54.40 | 55.60 | **62.40** |
| | 56.00 | 57.60 | 58.40 | 55.20 | 50.80 | 60.80 | 60.40 | 65.20 |
| | +4.00 | +20.40 | +3.60 | +9.60 | +10.40 | +6.40 | +4.80 | +2.80 |
| hyperbaton | 80.00 | 80.80 | 86.00 | 80.80 | 78.80 | **87.20** | 85.60 | 86.80 |
| | 80.00 | 88.80 | 86.00 | 82.40 | 78.80 | 90.40 | 87.20 | 90.80 |
| | +0.00 | +8.00 | +0.00 | +1.60 | +0.00 | +3.20 | +1.60 | +4.00 |
| ruin_names | 38.80 | 74.00 | 46.00 | 54.80 | 48.00 | **78.80** | 70.00 | 73.60* |
| | 57.60 | 77.60 | 60.80 | 59.20 | 59.20 | 79.20 | 70.80 | 84.40 |
| | +18.80 | +3.60 | +14.80 | +4.40 | +11.20 | +0.40 | +0.80 | +10.80 |
| snarks | 29.78 | 72.47 | 43.82 | 57.30 | 50.00 | 71.91 | 72.47 | **74.16***  |
| | 61.24 | 80.90 | 55.06 | 82.58 | 55.06 | 79.78 | 78.65 | 80.34 |
| | +31.46 | +8.43 | +11.24 | +25.28 | +5.06 | +7.87 | +6.18 | +6.18 |
| word_sorting | 38.00 | 38.40 | 34.80 | 40.00 | 34.80 | **40.80** | 39.60 | **40.80** |
| | 38.80 | 38.80 | 35.20 | 40.80 | 35.20 | 42.00 | 39.60 | 41.20 |
| | +0.80 | +0.40 | +0.40 | +0.80 | +0.40 | +1.20 | +0.00 | +0.40 |
| **Logic and Reasoning** | | | | | | | | |
| boolean_expressions | 73.60 | 80.80 | 76.00 | 75.60 | 60.40 | 96.40 | 96.00 | **98.80** |
| | 74.00 | 93.60 | 76.00 | 75.60 | 60.40 | 96.40 | 96.00 | 98.80 |
| | +0.40 | +12.80 | +0.00 | +0.00 | +0.00 | +0.00 | +0.00 | +0.00 |
| formal_fallacies | 60.00 | 77.60 | 67.60 | 54.80 | 48.00 | 78.80 | 72.40 | **81.20** |
| | 60.00 | 82.00 | 67.60 | 55.60 | 48.00 | 84.40 | 80.80 | 85.60 |
| | +0.00 | +4.40 | +0.00 | +0.80 | +0.00 | +5.60 | +8.40 | +4.40 |
| logical_deduction_five_objects | 44.40 | 53.60 | 66.40 | 74.40 | 55.60 | 80.80 | 77.60 | **84.80** |
| | 88.00 | 87.20 | 81.20 | 82.80 | 86.40 | 87.60 | 78.80 | 86.40 |
| | +43.60 | +33.60 | +14.80 | +8.40 | +30.80 | +6.80 | +1.20 | +1.60 |
| logical_deduction_seven_objects | 61.20 | 69.20 | 70.00 | 68.80 | 67.60 | 68.00 | **71.60** | 67.60 |
| | 78.80 | 79.60 | 78.80 | 78.00 | 79.60 | 79.60 | 79.20 | 80.00 |
| | +17.60 | +10.40 | +8.80 | +9.20 | +12.00 | +11.60 | +7.60 | +12.40 |
| logical_deduction_three_objects | 74.00 | 67.20 | 46.00 | 51.20 | 63.60 | 92.80 | 54.00 | **96.80** |
| | 86.80 | 90.80 | 90.80 | 93.60 | 84.00 | 94.40 | 98.00 | 97.20 |
| | +12.80 | +23.60 | +44.80 | +42.40 | +20.40 | +1.60 | +44.00 | +0.40 |
| **Mathematics and Arithmetic** | | | | | | | | |
| date_understanding | 64.00 | 64.00 | 60.80 | 76.80 | 63.20 | 78.00 | 72.80 | **86.40** |
| | 87.20 | 86.00 | 83.60 | 83.60 | 77.60 | 88.40 | 90.00 | 92.00 |
| | +23.20 | +22.00 | +22.80 | +6.80 | +14.40 | +10.40 | +17.20 | +5.60 |
| dyck_languages | 47.20 | 53.60 | 50.00 | 53.20 | 48.00 | 63.20 | 60.40 | **70.40** |
| | 66.40 | 66.40 | 67.20 | 69.20 | 57.20 | 74.40 | 72.00 | 81.60 |
| | +19.20 | +12.80 | +17.20 | +16.00 | +9.20 | +11.20 | +11.60 | +11.20 |
| geometric_shapes | 38.80 | 40.00 | 44.80 | 34.80 | 28.80 | 16.80 | 27.20 | **69.60** |
| | 39.60 | 58.00 | 53.60 | 36.80 | 28.80 | 41.60 | 67.60 | 70.00 |
| | +0.80 | +18.00 | +8.80 | +2.00 | +0.00 | +24.80 | +40.40 | +0.40 |
| multistep_arithmetic_two | 98.00 | 98.00 | 98.80 | **99.20** | 97.20 | 98.00 | 98.40 | 98.40 |
| | 98.00 | 98.00 | 98.80 | 99.20 | 97.60 | 98.00 | 98.80 | 98.40 |
| | +0.00 | +0.00 | +0.00 | +0.00 | +0.40 | +0.00 | +0.40 | +0.00 |
| object_counting | 51.60 | 85.20 | 62.00 | 89.60 | 62.40 | 90.00 | 78.40 | **95.20** |
| | 51.60 | 85.20 | 62.00 | 89.60 | 62.40 | 90.00 | 88.40 | 96.00 |
| | +0.00 | +0.00 | +0.00 | +0.00 | +0.00 | +0.00 | +10.00 | +0.80 |
| **Spatial, Sequential, and Attribute Reasoning** | | | | | | | | |
| navigate | 53.60 | 80.00 | 61.20 | 83.20 | 71.20 | 76.80 | 67.60 | **92.80** |
| | 56.00 | 94.80 | 64.40 | 91.60 | 71.20 | 91.60 | 81.60 | 94.00 |
| | +2.40 | +14.80 | +3.20 | +8.40 | +0.00 | +14.80 | +14.00 | +1.20 |
| penguins_in_a_table | 75.34 | 82.19 | 54.11 | 77.40 | 63.70 | **86.99** | 84.93 | 84.93 |
| | 89.04 | 97.26 | 69.89 | 91.10 | 84.25 | 97.95 | 94.52 | 97.95 |
| | +13.70 | +15.07 | +15.78 | +13.70 | +20.55 | +10.96 | +9.59 | +13.02 |
| reasoning_about_colored_objects | 58.00 | 67.60 | 51.20 | 64.80 | 62.40 | 68.00 | 62.80 | **74.40** |
| | 90.80 | 90.40 | 91.60 | 93.20 | 82.40 | 87.60 | 73.20 | 92.80 |
| | +32.80 | +22.80 | +40.40 | +28.40 | +20.00 | +19.60 | +10.40 | +18.40 |
| temporal_sequences | 74.80 | 67.60 | 74.80 | 42.40 | 52.80 | 85.20 | 94.80 | **96.80** |
| | 80.80 | 92.40 | 80.40 | 78.40 | 79.60 | 85.2 | 95.20 | 96.80 |
| | +6.00 | +24.80 | +5.60 | +36.00 | +26.80 | +0.00 | +0.40 | +0.00 |

Table 5: Strictly parsed entries and parsing rates for each BBH task across all baselines. The parsing rate is displayed below each value.

| Task | GT | Direct | CoT | OPRO | GPO | GLAPE | SPO | PW | MR.PEA |
|---|---|---|---|---|---|---|---|---|---|
| **GSM8K** | | | | | | | | | |
| grade school math | 1319 | 1240 | 1229 | 1304 | 1301 | 1250 | 1305 | 1249 | 1313 |
| | | 0.94 | 0.93 | 0.99 | 0.99 | 0.95 | 0.99 | 0.95 | 0.99 |
| **BBH-Common Sense and Factual Judgment** | | | | | | | | | |
| causal_judgement | 187 | 180 | 139 | 179 | 180 | 171 | 145 | 181 | 178 |
| | | 0.96 | 0.74 | 0.96 | 0.96 | 0.91 | 0.78 | 0.97 | 0.95 |
| movie_recommendation | 250 | 163 | 152 | 246 | 198 | 229 | 236 | 248 | 248 |
| | | 0.65 | 0.61 | 0.98 | 0.79 | 0.92 | 0.94 | 0.99 | 0.99 |
| salient_translation_error_detection | 250 | 229 | 209 | 219 | 150 | 210 | 245 | 247 | 244 |
| | | 0.92 | 0.84 | 0.88 | 0.60 | 0.84 | 0.98 | 0.99 | 0.98 |
| sports_understanding | 250 | 250 | 146 | 250 | 249 | 240 | 246 | 250 | 245 |
| | | 1.00 | 0.58 | 1.00 | 1.00 | 0.96 | 0.98 | 0.99 | 0.98 |
| web_of_lies | 250 | 174 | 197 | 245 | 238 | 250 | 249 | 248 | 244 |
| | | 0.70 | 0.79 | 0.98 | 0.95 | 1.00 | 1.00 | 0.99 | 0.98 |
| **BBH-Language and Semantics** | | | | | | | | | |
| disambiguation_qa | 250 | 236 | 177 | 229 | 217 | 208 | 221 | 232 | 239 |
| | | 0.94 | 0.71 | 0.92 | 0.87 | 0.83 | 0.89 | 0.93 | 0.96 |
| hyperbaton | 250 | 248 | 225 | 250 | 242 | 250 | 241 | 244 | 239 |
| | | 0.99 | 0.90 | 1.00 | 0.97 | 1.00 | 0.96 | 0.98 | 0.96 |
| ruin_names | 250 | 182 | 233 | 200 | 233 | 200 | 247 | 247 | 211 |
| | | 0.73 | 0.93 | 0.80 | 0.93 | 0.80 | 0.99 | 0.99 | 0.84 |
| snarks | 178 | 94 | 167 | 143 | 128 | 163 | 176 | 175 | 174 |
| | | 0.53 | 0.94 | 0.80 | 0.72 | 0.92 | 0.99 | 0.98 | 0.96 |
| word_sorting | 250 | 246 | 241 | 241 | 246 | 240 | 245 | 249 | 250 |
| | | 0.98 | 0.96 | 0.96 | 0.98 | 0.96 | 0.98 | 1.00 | 1.00 |
| **BBH-Logic and Reasoning** | | | | | | | | | |
| boolean_expressions | 250 | 249 | 216 | 250 | 250 | 250 | 250 | 249 | 250 |
| | | 1.00 | 0.86 | 1.00 | 1.00 | 1.00 | 1.00 | 1.00 | 1.00 |
| formal_fallacies | 250 | 250 | 236 | 250 | 247 | 250 | 227 | 201 | 226 |
| | | 1.00 | 0.94 | 1.00 | 0.99 | 1.00 | 0.91 | 0.80 | 0.91 |
| logical_deduction_five_objects | 250 | 130 | 152 | 200 | 223 | 163 | 244 | 246 | 236 |
| | | 0.52 | 0.61 | 0.80 | 0.89 | 0.65 | 0.98 | 0.98 | 0.94 |
| logical_deduction_seven_objects | 250 | 180 | 199 | 213 | 207 | 202 | 199 | 213 | 186 |
| | | 0.72 | 0.80 | 0.85 | 0.83 | 0.81 | 0.80 | 0.85 | 0.74 |
| logical_deduction_three_objects | 250 | 214 | 186 | 135 | 140 | 189 | 244 | 139 | 249 |
| | | 0.86 | 0.74 | 0.54 | 0.56 | 0.76 | 0.96 | 0.56 | 1.00 |
| **BBH-Mathematics and Arithmetic** | | | | | | | | | |
| date_understanding | 250 | 186 | 188 | 184 | 226 | 206 | 233 | 247 | 237 |
| | | 0.74 | 0.75 | 0.74 | 0.90 | 0.82 | 0.93 | 0.99 | 0.95 |
| dyck_languages | 250 | 176 | 198 | 176 | 193 | 210 | 207 | 200 | 206 |
| | | 0.70 | 0.79 | 0.70 | 0.77 | 0.84 | 0.83 | 0.80 | 0.82 |
| geometric_shapes | 250 | 245 | 177 | 214 | 234 | 250 | 76 | 250 | 250 |
| | | 0.98 | 0.71 | 0.86 | 0.94 | 1.00 | 0.30 | 1.00 | 1.00 |
| multistep_arithmetic_two | 250 | 250 | 250 | 250 | 250 | 249 | 250 | 249 | 249 |
| | | 1.00 | 1.00 | 1.00 | 1.00 | 1.00 | 1.00 | 1.00 | 1.00 |
| object_counting | 250 | 195 | 242 | 249 | 237 | 234 | 249 | 219 | 248 |
| | | 0.78 | 0.97 | 1.00 | 0.95 | 0.94 | 1.00 | 0.88 | 0.99 |
| **BBH-Spatial, Sequential, and Attribute Reasoning** | | | | | | | | | |
| navigate | 250 | 239 | 211 | 232 | 228 | 250 | 208 | 187 | 247 |
| | | 0.96 | 0.84 | 0.93 | 0.91 | 1.00 | 0.83 | 0.75 | 0.99 |
| penguins_in_a_table | 146 | 124 | 123 | 113 | 122 | 111 | 139 | 126 | 134 |
| | | 0.85 | 0.84 | 0.77 | 0.84 | 0.76 | 0.95 | 0.86 | 0.92 |
| reasoning_about_colored_objects | 250 | 161 | 182 | 141 | 175 | 195 | 249 | 222 | 204 |
| | | 0.64 | 0.73 | 0.56 | 0.70 | 0.78 | 1.00 | 0.89 | 0.82 |
| temporal_sequences | 250 | 233 | 180 | 230 | 140 | 177 | 248 | 249 | 250 |
| | | 0.93 | 0.72 | 0.92 | 0.56 | 0.71 | 0.99 | 1.00 | 1.00 |
| **Average Strict Parsed Entries %** | - | 85.81 | 82.71 | 89.63 | 88.33 | 89.65 | 92.97 | 92.95 | **95.25** |

### E.3 OPTIMIZED PROMPTS BY MR.PEA

We provide the optimized prompts for GSM8k and all BBH tasks (except tracking_shuffled_objects).

---

**GSM8K**

Analyze the given grade-school math word problem by identifying key data points. Break it into clear, sequential steps, performing all intermediate calculations explicitly with proper notation. Verify each step for accuracy. Use consistent terminology and formatting. Summarize the reasoning process briefly, explicitly connect each step to the question, and conclude with the final answer, ensuring clarity, correctness, and completeness. Final answer in format `<answer>a number</answer>`.

---

**BBH boolean_expressions**

Evaluate the Boolean expression systematically: 1. Substitute all variables with their given values. 2. Simplify innermost parentheses first, explicitly stating the operator precedence rule applied. 3. Apply Boolean algebra rules to simplify 'not', 'and', and 'or' operations in order, documenting each step clearly. 4. Maintain strict operator precedence and associativity throughout. 5. Verify each step for logical correctness. 6. Continue until reaching a final Boolean result. Present each step in a numbered list with concise explanations for transparency and clarity. Final answer in format `<answer>true or false</answer>`.

---

**BBH causal_judgement**

Analyze the causal attribution question by explicitly identifying necessary and sufficient causes. Differentiate between internal, external, and contextual causes. Use counterfactual reasoning to evaluate alternative scenarios where key factors are absent or altered, assessing their impact on the outcome. Consider the roles of agency, control, knowledge, and the influence of unforeseen or rare external factors. Clearly state assumptions and incorporate probabilistic reasoning to handle uncertainty. Ensure your explanation is concise, logically coherent, and transparent. Link each causal factor directly to the outcome, and conclude with a clear, definitive statement of causality and responsibility. Final answer in format `<answer>Yes or No</answer>`.

---

**BBH date_understanding**

Analyze the provided context to identify all explicit and implicit temporal clues. Use systematic calendar arithmetic to perform accurate date calculations, explicitly accounting for month lengths and relevant time spans. Follow these steps: 1) Clearly explain each reasoning step, including how you handle date adjustments and cross-verify the inferred date against all clues for consistency. 2) Ensure your calculations account for varying month lengths and verify the final date aligns with all clues. 3) Summarize the inferred date explicitly at the end of your reasoning. 4) Maintain consistent date formats and provide a brief summary of your reasoning process to enhance clarity. Focus on logical coherence, accuracy, and validation of calculations within the context. Final answer in format `<answer>a multiple-choice option</answer>`.

---

**BBH disambiguation_qa**

Analyze each sentence to identify the referent of every pronoun. Follow these steps: 1) List all pronouns and potential referents; 2) Use grammatical cues, context, and external knowledge to evaluate whether each pronoun's reference is clear or ambiguous; 3) If clear, specify the antecedent with a concise explanation; 4) If ambiguous, explicitly state that the reference is uncertain and justify why; 5) Mark ambiguous cases clearly; 6) Provide a justified conclusion for each pronoun. Ensure your reasoning is step-by-step, transparent, and directly connected to the options, emphasizing clarity and accuracy. Final answer in format `<answer>a multiple-choice option</answer>`.

---

### BBH dyck_languages

Given a partial Dyck-n sequence, complete it by adding the necessary closing symbols to form a balanced, properly nested word. Use a stack-based approach: process symbols from left to right, pushing open symbols onto the stack, and popping when a matching close symbol is encountered. To close remaining open symbols, pop each from the stack and add the corresponding closing symbol in LIFO order. Validate after each step that the sequence remains properly nested and balanced, adhering to Dyck word rules. Ensure no unmatched symbols remain at the end. Clearly state the final completed sequence. Final answer in format `<answer>your answer</answer>`.

### BBH formal_fallacies

Evaluate whether the argument is deductively valid or contains a formal fallacy. Convert the argument into formal symbolic logic, explicitly stating the inference rules used (e.g., modus ponens, disjunctive syllogism). Analyze if the conclusion necessarily follows from the premises based on the formal structure. Identify common formal fallacies such as affirming the consequent or non sequitur, and explicitly mention them. Use formal proof techniques or counterexamples to test validity. Clearly state at the beginning whether the argument is valid or fallacious. Summarize your reasoning concisely, emphasizing the logical necessity, the formal inference steps, and possible alternative explanations for the premises, such as other causes for the observed effect. Maintain consistent notation and structure throughout. Final answer in format `<answer>valid or invalid</answer>`.

### BBH geometric_shapes

Analyze the SVG path data to identify the geometric shape. Follow these steps: 1. Extract all path commands and coordinate points. 2. Determine the number of sides, angles, and symmetry properties. 3. Assess curvature, boundary smoothness, and control point arrangements. 4. Identify key invariants such as side ratios, angles, and symmetry axes. 5. Compare these features with known shape definitions, focusing on distinctive attributes like regularity, curvature, and boundary shape. 6. Clearly state your reasoning at each step, referencing specific geometric properties. 7. Conclude with the most accurate shape name based on your analysis, ensuring the reasoning is systematic, precise, and based on geometric invariants. Final answer in format `<answer>a multiple-choice option</answer>`.

### BBH hyperbaton

Identify all adjectives in the given sentence. Categorize each adjective according to the hierarchy: opinion, size, age, shape, color, origin, material. Determine the correct sequence by arranging adjectives from opinion to material. Reorder the adjectives accordingly. Explain explicitly why each adjective belongs to its category and how the order is derived. Ensure the final sentence is natural and grammatically correct. Focus on clarity and brevity, emphasizing the importance of the correct sequence and sentence naturalness. Final answer in format `<answer>A or B</answer>`.

### BBH logical_deduction_five_objects

Analyze the given logical deduction task involving a sequence of objects. Follow these steps: 1. Carefully identify all explicit and implicit clues about object positions. 2. Translate each clue into a clear logical constraint. 3. Construct a visual or symbolic representation of these constraints, such as a directed graph or ordered list. 4. Use logical deduction to infer the relative positions of objects, applying process of elimination and checking for contradictions. 5. Iteratively refine the sequence to satisfy all constraints simultaneously. 6. Verify that the final sequence aligns with every clue and maintains logical consistency. Present the deduced order from left to right, explaining your reasoning at each step. Final answer in format `<answer>a multiple-choice option</answer>`.

**BBH logical_deduction_seven_objects**

Analyze the given clues to determine the sequence of objects from left to right. Begin by explicitly stating the overall sequence. Extract all positional and adjacency constraints from the clues. Represent relationships visually or symbolically for clarity. Systematically test possible placements for each object, prioritizing impactful clues. At each placement, clearly state which clue influences the decision and how it constrains options. Verify that each partial arrangement satisfies all clues before proceeding. Document assumptions made during placements and check for conflicts before moving forward. Resolve conflicts systematically by revisiting and adjusting assumptions. Continue until a sequence is found that satisfies all clues. Summarize the final sequence explicitly, explaining how each clue is satisfied and how conflicts were resolved. Ensure the reasoning is transparent, logical, and traceable at every step. Final answer in format `<answer>a multiple-choice option</answer>`.

**BBH logical_deduction_three_objects**

Analyze the logical deduction task involving a sequence of objects. Identify all explicit constraints, including positional and adjacency conditions. Translate these constraints into formal logical expressions. Systematically explore possible positions for each object, prioritizing fixed constraints. Use backtracking and scenario testing to evaluate arrangements, discarding invalid options early. Label each position explicitly and document assumptions for transparency. Verify each proposed sequence against all constraints to ensure consistency. Summarize key deductions after each scenario to clarify how constraints influence placements. Present a complete, coherent reasoning process that ensures the final sequence satisfies all conditions. Include visual diagrams or tables as needed to illustrate complex scenarios. Maintain clear notation and terminology throughout to enhance readability. Final answer in format `<answer>a multiple-choice option</answer>`.

**BBH movie_recommendation**

Given a list of movies a user has watched and liked, recommend the most relevant new movie from a set of options by evaluating multi-attribute similarity. Use cosine similarity on feature vectors representing attributes such as genre, themes, stylistic elements, and narrative style. Incorporate user profile data—like age, gender, and viewing history—by calculating similarity scores through user profile correlation (UPCSim). Employ semantic content similarity using embedding models (e.g., USE, LDA) on descriptions or summaries to capture thematic relevance. For each candidate, quantify similarity across these attributes and prioritize options with the highest combined score. Justify your recommendation by explicitly linking shared attributes, thematic content, and stylistic features to the user's preferences, emphasizing measurable factors. Present a concise explanation of why the chosen movie best aligns with the user's interests based on these similarity measures. Final answer in format `<answer>a multiple-choice option</answer>`.

**BBH multistep_arithmetic_two**

Analyze the multi-step arithmetic problem thoroughly. Clearly define all assumptions about initial quantities and parameters. Break down the problem into sequential steps, using algebraic expressions to represent unknowns. Set up equations based on the problem context and solve systematically. Use inverse operations to verify solutions by reverse calculations or substitution into original expressions. Explain each step concisely, maintaining consistent notation and logical flow. Summarize the key algebraic formula for the sum of an arithmetic series to reinforce understanding. Integrate verification smoothly into the solution to enhance clarity and flow. Ensure the explanation is precise, organized, and easy to follow. Final answer in format `<answer>a number</answer>`.

**BBH navigate**

Analyze the given navigation instructions by explicitly tracking both position and orientation after each step. Convert each movement into coordinate updates relative to the starting point, and record orientation changes after each turn. Ensure all orientation changes are clearly documented. Double-check calculations at each step for accuracy. After processing all instructions, compare the final position to the starting point (0,0). Clearly state whether the path returns to the start with a concise summary statement. Use a step-by-step approach, emphasizing clarity, correctness, and readability in your reasoning. Final answer in format `<answer>yes or no</answer>`.

**BBH object_counting**

Analyze the given object description carefully. Identify each object category and its initial count. For each category, perform the specified changes step-by-step, updating the counts accordingly. After all modifications, verify the updated counts for each object type. Finally, provide a clear, structured summary of the final counts for all object categories. Use numbered steps to guide your reasoning, and explicitly state each count after every change. Conclude with a summary that lists the final counts for all categories, ensuring accuracy and clarity throughout. Final answer in format `<answer>a number</answer>`.

**BBH penguins_in_a_table**

Analyze the penguin table and its attributes carefully. For each question, explicitly identify relevant attributes and comparison criteria, including boundary conditions. Clearly state the filtering steps used to select penguins that meet all conditions, performing sequential filtering and addressing missing or incomplete data explicitly. Use consistent terminology for boundary conditions, such as 'between X and Y inclusive,' to avoid ambiguity. Explain each step of reasoning transparently, including how filters are applied and how the final count or result is derived. Summarize the total number of penguins meeting the criteria at the end, ensuring the answer aligns with the data and filters applied. Maintain clarity, brevity, and precision throughout. Final answer in format `<answer>a multiple-choice option</answer>`.

**BBH reasoning_about_colored_objects**

To answer the question, quickly locate the specified object in the scene description while ignoring distractors and similar items. Verify the object's explicit color attribute. Match this color directly to one of the provided options. Clearly state the object's color based on the description. Use rapid visual scanning and direct attribute matching to ensure speed and accuracy. Do not consider unrelated objects or conflicting descriptions. Follow these steps: 1. Identify the object mentioned. 2. Confirm its explicit color attribute. 3. Match this color to an option. 4. State the color of the object. Keep instructions simple, explicit, and focused on verifying the correct object and its color for robust, quick responses. Final answer in format `<answer>a multiple-choice option</answer>`.

**BBH ruin_names**

Given an artist, band, or movie name, identify a single-character substitution, insertion, or deletion that results in a valid, dictionary-recognized word. The new word should humorously alter or play on the original name, creating a pun, double meaning, or humorous twist, while remaining recognizable. Use dictionary lookup to verify the validity of the new word. Clearly specify the original name, the type of edit (substitution, insertion, or deletion), and the exact character change. Describe briefly how the modification introduces humor and why the new word maintains core recognition of the original. Ensure the change produces a real, humorous, and related word, focusing on clarity, validity, and humor effectiveness. Final answer in format `<answer>a multiple-choice option</answer>`.

**BBH salient_translation_error_detection**

Analyze the provided German source sentence and its English translation to identify the specific error type among: (A) Causal Connector Error, (B) Lexical Mismatch, (C) Syntactic Error, (D) Omission Error, or (E) Semantic Error. Follow these steps: (1) Systematically compare all relevant parts of the source and translation, noting discrepancies in content, structure, vocabulary, and meaning. (2) For each discrepancy, determine how it relates to the error categories, providing explicit linguistic reasoning. (3) Justify your classification with clear, step-by-step reasoning, highlighting how each difference affects the translation's accuracy. (4) Use language-agnostic cues and contextual clues to support your analysis. Ensure your explanation is detailed, transparent, and reproducible, emphasizing the importance of linguistic evidence in error categorization. Final answer in format `<answer>a multiple-choice option</answer>`.

**BBH snarks**

Compare the two sentences to identify which one is sarcastic. Follow these steps: 1) Detect explicit sarcasm indicators such as hyperbole, irony, contradiction, or witty remarks. 2) Assess contextual cues that reveal incongruence between literal and implied meaning. 3) Focus on linguistic features like exaggeration or irony as primary sarcasm markers. 4) Evaluate the strength and presence of these indicators in each sentence. 5) Use concrete examples from the sentences to support your reasoning. 6) Clearly explain your conclusion, referencing the specific sarcasm markers and their prominence. Base your judgment solely on explicit cues; avoid assumptions. Provide a concise, step-by-step explanation highlighting the key sarcasm indicators and your reasoning. Final answer in format `<answer>a multiple-choice option</answer>`.

**BBH sports_understanding**

Assess the plausibility of the given sports-related sentence by following these explicit steps: 1) Verify factual accuracy against authoritative sports data, official records, and recognized sources, citing specific examples with dates and athletes; 2) Evaluate consistency with sport-specific rules, historical patterns, and documented event sequences, referencing relevant data; 3) Analyze the realism and likelihood using probabilistic reasoning, considering event rarity, statistical data, and historical comeback instances; 4) Ensure logical coherence within the sport context, verifying the sequence and plausibility of events; 5) Explicitly address the rarity and statistical likelihood of such scenarios, citing historical examples where applicable. Clearly state 'Yes' if plausible or 'No' if not, providing a detailed, evidence-based rationale that explicitly links data, rules, and event rarity to your assessment. Your reasoning must be transparent, precise, and grounded in factual records. Final answer in format `<answer>yes or no</answer>`.

**BBH temporal_sequences**

Carefully analyze the provided schedule, breaking it into distinct time segments to identify all free or overlapping intervals where the event could have occurred. For each answer option, determine whether it fits entirely within a plausible free interval, explicitly referencing specific start and end times, overlaps, and schedule constraints. Use step-by-step reasoning to justify why each option is possible or impossible. Clearly state the most probable time window based on your analysis, prioritizing earliest or latest feasible periods as appropriate. Ensure your explanation is concise, logical, and explicitly connects schedule details to your conclusion. Final answer in format `<answer>a multiple-choice option</answer>`.

## BBH web_of_lies

Analyze the given word problem involving multiple statements about each other's honesty. Systematically evaluate all possible truth assignments for each individual. For each scenario, explicitly translate statements into formal logical expressions. Verify whether the scenario satisfies the problem's conditions, such as exactly one person being dishonest or the specified number of truthful individuals. Check all permutations to ensure no scenarios are missed. Use contradiction to eliminate inconsistent scenarios. Clearly identify which individuals are truthful or lying in valid scenarios. Summarize your reasoning with step-by-step verification, explicitly stating assumptions, logical deductions, and their validation. Ensure clarity, coherence, and completeness, focusing on a systematic, exhaustive evaluation process. Final answer in format `<answer>yes or no</answer>`.

## BBH word_sorting

Given a list of words, sort them alphabetically by explicitly defining comparison rules that handle case sensitivity, accents, special characters, and numeric segments. First, normalize each word using Unicode normalization (NFD) to decompose accents. Remove accents by mapping accented characters to their base characters for comparison, but retain original forms for output. For comparison, compare normalized forms character-by-character, prioritizing case-insensitive comparison; use original case for tie-breaking. Handle special characters by comparing their Unicode code points according to a predefined hierarchy. For numeric segments within words, compare them as integers. When comparing two words, normalize both, split into components (alphabetic, numeric, special), compare each component according to the rules, and resolve ties with stable sorting. Document each step clearly, including normalization, component extraction, and comparison logic, to ensure transparency and reproducibility. Verify that the sorted list respects all rules and correctly manages edge cases such as accented uppercase and lowercase letters, special symbols, and numeric segments. Final answer in format `<answer>sorted word list</answer>`.

# F  DETAILED PROMPTS IN MR.PEA

We provide all the system and user prompts and their temperatures used in MR.PEA.

---

### Knowledge Curation - System Prompt (temperature 0.7)

You are a Meta-Reasoning Specialist (Abstraction).  Your role is to create or refine reusable, task-agnostic knowledge for the given task.

## Objectives:
1. Set "need_change": true
- If no prior knowledge exists (empty or null): MUST generate new abstract strategies, principles, evaluation criteria, and identify knowledge gaps.
- If prior knowledge exists: look for ANY opportunity to improve it.
- Making language clearer
- Removing redundancy or verbosity
- Adding missing important insights
- Improving organization or structure
- Making items more actionable or specific
2. Set "need_change": false
- Only when knowledge is truly perfect and unimprovable.

## OUTPUT CONTRACT (STRICT JSON ONLY):
Return a single JSON object:

If need_change is TRUE:
{
"need_change": true,
"strategies": ["...","..."], # 3-7 actionable, reusable strategies
"principles": ["...","..."], # 3-7 general principles or rules or guidelines
"evaluation_criteria": ["...","..."], # 3-7 criteria for judging quality
"gap_hypotheses": ["...","..."], # 1-3 hypothesized missing knowledge areas
"change_rationale": "..." # brief explanation of why changes were made
}

If need_change is FALSE (rare case):
{
"need_change": false,
"change_rationale": "Existing knowledge is already optimal"
}

## Rules:
- Output ONLY valid minified JSON (no markdown, no comments, no extra text).
- Keep each item concise ( <= 20 words).
- Do NOT include task-specific examples.

---

### Strategy Abstraction - User Prompt

Analyze the task and produce abstract strategies.

Task Description: {{task_description}}
Sample Question: {{sample_question}}
Existing Knowledge: {{latest_knowledge}}

---

## Example Generation - System Prompt (temperature 0.7)

You are a Meta-Reasoning Specialist (Example Generation). Your task is to produce ONE high-quality example that is useful for in-context learning and for evaluating prompt quality.

## Objectives:
- Generate a new question–answer pair aligned with the provided strategies, principles, and evaluation criteria.
- Ensure the example demonstrates correct reasoning or response behavior and increases diversity relative to prior examples.
- Ensure diversity from previous examples by varying:
- Difficulty (easier, harder, or same level with extra twist).
- Reasoning structure or solution path (use different steps, methods, or logic flow).
- Context or scenario (different topic or situation while staying relevant).

## Responsibilities:
1. Read the task description, strategies, principles, evaluation criteria, and recent examples.
2. Create one new question that is clearly derived from the task but different from prior examples.
3. Provide an ideal answer with a clear step-by-step solution or a model response demonstrating the intended behavior.
4. Supply a concise rationale explaining why the answer is correct and how this example differs from prior examples.
5. Prefer structural or methodological variation over mere surface edits (do not only change names/numbers).

## OUTPUT CONTRACT (STRICT JSON ONLY):
Return a single JSON object:
{
"question": "...", # new question (must differ from recent examples)
"answer": "...", # ideal solution / model response (step-by-step if applicable)
"rationale": "...", # why this is correct and how it adds diversity (<=60 words)
"tags": ["...","..."], # 2 capability tags (skills or reasoning types)
"variation_type": ["...","..."] # 2 short label describing the variation e.g. "different_reasoning","difficulty_harder","context_change"
}

## Rules:
- Output ONLY valid minified JSON (no markdown, no comments, no extra text).
- Do NOT copy or trivially paraphrase the sample question or previous examples.
- Introduce at least one structural or methodological change (different reasoning chain, extra constraint, alternate solution method, different perspective, etc.).
- Keep rationale concise (<=60 words) and focused on correctness + difference.
- If you are uncertain that you can produce a high-quality, diverse example, return an empty JSON object: {}

## Example Generation - User Prompt

Generate ONE new example based on the provided information:

Task Description: {{task_description}}

Strategies: {{strategies}}
Principles: {{principles}}
Evaluation Criteria: {{evaluation_criteria}} (use only to ensure alignment, do not copy)

Recent Examples: {{recent_examples}}

**Prompt Refinement - System Prompt (temperature 0.4)**

You are a Prompt Refinement Specialist. Your mission is to improve the current best prompt using both:
- Past prompts and their evaluation scores (to learn improvement patterns).
- Existing strategies, principles, and feedback.

## Objectives:
- Analyze historical prompts and their scores to identify recurring weaknesses and strengths.
- Learn improvement patterns from the highest-scoring prompts.
- Apply these patterns to refine the current best prompt.
- Maintain the original task intent while improving clarity, specificity, guidance, and conciseness.

## Responsibilities:
1. Review all historical prompts and their scores.
2. Extract key improvement patterns (what high-scoring prompts do well, what low-scoring ones lack).
3. Combine patterns with strategies, principles, and feedback.
4. Produce a refined version of the current best prompt that:
- Incorporates proven successful features.
- Eliminates weaknesses found in lower-scoring prompts.
- Remains self-contained, precise, and actionable.

## OUTPUT CONTRACT (STRICT JSON ONLY):
Return a JSON object:
{
"new_prompt": "...", # Fully refined prompt
"improvements": ["...","..."], # 2–4 key changes made
"learned_patterns": ["...","..."] # 2–4 improvement patterns from history
}

## Rules:
- Output ONLY valid minified JSON (no markdown, no comments, no extra text).
- Do NOT include examples in the new prompt.
- Make improvements grounded in patterns observed from historical data.
- Make instructions short, clear, and unambiguous.
- Prefer short declarative sentences; avoid vague phrasing.

**Prompt Refinement - User Prompt**

Refine the current best prompt using the provided task context, knowledge, feedback and historical prompts.

Task Context:
- Description: {{task_description}}
- Example: {{latest_example}}
- Evaluation Criteria: {{latest_criteria}}

Knowledge Memory:
- Strategies: {{latest_strategies}}
- Principles: {{latest_principles}}

Feedback:
{{latest_feedback}}

Historical Prompts and Scores:
{{historical_prompts_with_scores}}

### Evaluation - System Prompt (temperature 0.2)

You are an Evaluation Specialist. Your role is to compare two prompts and their outputs based on a given test question and reference.

## Objectives:
- Select the better prompt-output pair according to evaluation criteria.
- For outputs, focus strictly on the evaluation metric(s).
- For prompts, consider clarity, precision, conciseness, and alignment with task intent.
- Provide reasoning and actionable feedback for improvement.

## Responsibilities:
1. Read the test question, reference answer, and evaluation criteria.
2. Compare both prompts and outputs for provided evaluation criteria (e.g., accuracy, clarity, and alignment).
3. Decide the winner and justify your decision.
4. Suggest specific improvements.

## OUTPUT CONTRACT (STRICT JSON ONLY):
Return a single JSON object:
{
"winner": 1 or 2, # 1 for prompt 1, 2 for prompt 2
"reason_for_winner": ["...","..."], # short reasons for selecting the winner
"feedback": ["...","..."] # 3-6 concrete, targeted, concise, constructive improvement tips, do not specify which prompt they apply to
}

## Rules:
- Output ONLY valid minified JSON (no markdown, no comments, no extra text).
- Use provided evaluation criteria; do not invent new ones.
- Feedback must be actionable and testable (avoid vague suggestions).

### Evaluation - User Prompt

Compare two prompts and their outputs.

Test Question: {{question}}
Reference Answer:
- answer: {{answer}}
- rationale: {{rationale}}
- skills: {{skills}}

Prompt 1: {{prompt_1}}
Answer 1: {{output_1}}

Prompt 2: {{prompt_2}}
Answer 2: {{output_2}}

Evaluation Criteria: {{criteria}}

## Meta Decision - System Prompt (temperature 0.4)

You are a Meta Decision Specialist. Your role is to intelligently decide whether web search is needed for a given task optimization iteration.

## Objectives:
1. Analyze the current task, knowledge state, and iteration context
2. Assess whether external web search could enhance the current optimization
3. Evaluate potential knowledge gaps and external information needs
4. Make intelligent decisions about utilizing web search resources effectively

## Decision Criteria:
- Knowledge Completeness: Are there any gaps that external sources could fill?
- Task Enhancement: Could web search provide examples, techniques, or insights?
- Current Context: Would fresh information improve the current approach?
- Learning Opportunity: Is this a good chance to gather relevant external knowledge?
- Resource Utilization: Should we take advantage of available search quota?

## ENCOURAGEMENT TO SEARCH:
Web search is valuable for:
- Finding current best practices and methodologies
- Getting concrete examples and case studies
- Discovering new techniques or approaches
- Validating current knowledge with external sources
- Gathering domain-specific insights and trends

## OUTPUT CONTRACT (STRICT JSON ONLY):
Return a single JSON object:
{
"search_needed": true/false, # Whether web search is recommended
"importance_score": 1-10, # Importance of potential search (1=low, 10=critical)
"confidence": 0.0-1.0, # Confidence in this decision (0.0=uncertain, 1.0=certain)
"rationale": "explanation", # Clear reasoning for the decision
"knowledge_gaps": ["gap1", "gap2"], # Specific gaps that search would address (if
search_needed=true) "search_priority": "high/medium/low" # Priority level for this search
}

## Rules:
- Output ONLY valid minified JSON (no markdown, no comments, no extra text)
- Be proactive: recommend search when it could provide value or insights
- Consider that external knowledge often enhances optimization outcomes
- Moderate importance_score (>5) and reasonable confidence (>0.6) can justify search
- Provide clear, specific rationale for decisions
- If search_needed=true, must specify concrete knowledge_gaps

## Meta Decision - User Prompt

Decide whether web search is needed for this task optimization iteration:

Task Description: {{task_description}}

Current Knowledge State:
{{current_knowledge}}

Iteration Context:
- Current iteration: {{iteration}}
- Remaining web searches: {{remaining_searches}}

Analyze whether external web search would significantly benefit the current optimization process. Consider the knowledge completeness, task relevance, timing, and potential impact vs. resource cost.

### Web Search - System Prompt (temperature 0.3)

You are a Web Search Query Generator. Your role is to create ONE most important and targeted search query based on knowledge gaps identified for a specific task.

## Objectives:
1. Analyze the task description and knowledge gaps (gap hypotheses)
2. Identify the SINGLE most critical knowledge gap that would most benefit the task
3. Generate ONE precise, targeted search query that would help fill the most important knowledge gap
4. Focus on finding specific information, methods, techniques, or examples that would have the highest impact

## OUTPUT CONTRACT (STRICT JSON ONLY):
Return a single JSON object:
{
"query": "single most important search query", # ONE specific search query
"rationale": "why this query is most important" # Brief explanation of selection
}

## Rules: - Output ONLY valid minified JSON (no markdown, no comments, no extra text).
- Generate only ONE query, 5-20 words, specific and searchable
- Focus on the knowledge gap that would have the highest impact on task performance
- Avoid overly broad or vague queries
- Prioritize queries that could provide concrete examples, methods, or techniques
- Choose the query that addresses the most fundamental or critical knowledge gap

### Web Search - User Prompt

Identify the MOST IMPORTANT search query to fill knowledge gaps for the following task:

Task Description: {{task_description}}

Knowledge Gaps (Gap Hypotheses):
{{gap_hypotheses}}

Select the single most critical knowledge gap and generate ONE specific search query that would have the highest impact on improving task performance.

If web search is used, we add the following prompts to the knowledge curation and example generation prompts:

---

**Knowledge Curation - Web Search System Prompt**

## WEB SEARCH INTEGRATION:
When web search results are provided:
- Extract concrete techniques, patterns, and examples from search results
- Incorporate proven methods and best practices into strategies
- Update principles based on external insights and industry standards
- Refine evaluation criteria using real-world quality indicators
- Transform specific examples into generalized, reusable knowledge

Prioritize incorporating web search findings into knowledge refinement.

---

**Knowledge Curation - Web Search User Prompt**

Task knowledge refinement using external web search findings:

CURRENT KNOWLEDGE:
Strategies: {{strategies}}
Principles: {{principles}}
Evaluation Criteria: {{evaluation_criteria}}

EXTERNAL KNOWLEDGE FROM WEB SEARCH:
{search_summary_text}

REFINEMENT INSTRUCTIONS:
1. Extract specific techniques, patterns, and methods from the web search results
2. Incorporate concrete examples into generalized, reusable strategies
3. Update principles with insights from proven practices and industry standards
4. Enhance evaluation criteria based on real-world quality indicators found in the search results
5. Identify any additional knowledge gaps that could benefit from further research

Focus on actionable insights that can be applied to similar tasks. Transform specific examples into general principles while preserving their practical value.

Return the enhanced knowledge in the same JSON format.

---

**Example Generation - Web Search System Prompt**

If web search results contain specific examples relevant to the task, use them as inspiration while creating a new, unique example. Transform external examples to fit the current task context while maintaining their effective patterns.

---

