# OpenReview forum: "MR.PEA: The Meta-Reasoning Prompt Engineering Agent"
_ICLR.cc/2026/Conference — Submitted to ICLR 2026_

### Official Review · Reviewer_XDMT · 2025-10-25

**Soundness:** 3
**Presentation:** 3
**Contribution:** 2
**Rating:** 4
**Confidence:** 4

**Summary:**

The paper presents a prompt optimization framework (MR.PEA) featuring a meta-reasoning approach that builds task-specific knowledge for prompt optimization. The method has 5 main modules which covers various aspects and challenges in prompt optimization.
The method achieves strong empirical results on GSM8k and BIG-bench hard tasks.
Further the method uses fewer tokens and thus saves the optimization cost compared to other methods.
The paper also conducts ablation study to justify each components in the proposed framework.

**Strengths:**

* Realistic problem setting. Prompt optimization with limited data under budget constraint is a realistic problem with a lot of use cases.
* Comprehensive experiments. The authors mentioned reimplementing the baseline methods to ensure fair comparison, which is a lot of efforts and should be encouraged.
* Comprehensive ablation. The authors conducted detailed ablation on each module of the proposed method.

**Weaknesses:**

* Unclear motivation for using meta-reasoning: The introduction begins by introducing limitations of prior methods, but it's unclear why would meta-reasoning address these limitations (e.g., reliance on labeled data, high computation costs). I still see gaps between these limitations and the proposed method.
* Unclear distinction between "meta-reasoning" and "reasoning". The authors defined "meta-reasoning" as "the ability to reflect on and improve one's reasoning process". Is this mainly referring to phase 1 in your approach? What makes it "meta"? Which part of the reasoning is improved after applying your method?

**Questions:**

* Are all 5 phases in the method newly proposed in this paper, or were some of them used before in prior works? I recommend providing a table to highlight the distinctions from prior works.
* How are supervised prompt optimization methods used in Table 1? Do you apply them directly in the self-supervised setting with only one training example?
* Since MR.PEA introduces a new meta-reasoning centric module compared to prior work, I'm unsure why MR.PEA is cost-effective compared to the baseline methods. Could you please further clarify? Why are fewer tokens needed comparing to the baselines?
* Do you control the prompt lengths in your experiments? In the example in figure 2, it seems that the prompt generated by MR.PEA is significantly longer than the other two, which could give this method some advantages.
* I'm a bit confused at the terminology "meta-reasoning knowledge"/"knowledge" (e.g., line 198/419). If it's referring to strategies like "break down complex problems into smaller, manageable steps", this seems to be part of the final prompt of GSM8k ("Break it into
clear, sequential steps..."). How is curating knowledge (the problem solving strategy part) different from writing the final prompt?

* I feel the following work is highly related. Could the authors explain the relevance and differences with this work? https://proceedings.neurips.cc/paper_files/paper/2024/hash/e41efb03e20ca3c231940a3c6917ef6f-Abstract-Conference.html

---

> ### Author Response · Authors · 2025-11-21
> **Response to Reviewer XDMT (1/4)**
>
> We thank the reviewer XDMT for the thoughtful feedback. We appreciate your recognition of our practical problem setting and comprehensive empirical study. Below, we address your concerns regarding the motivation for meta-reasoning, distinctions from prior work, cost-effectiveness and other clarification.
>
> ## Motivation
>
> Our method eliminates the reliance on labeled data by adopting a self-supervised approach, where prompt optimization begins with only an unlabeled data sample and a concise task description. The introduction of "meta-reasoning" is to address two key limitations of existing self-supervised prompt optimization methods:
>
> 1. High computation cost for finding a reliable optimization direction: Methods such as GLaPE rely on consistency-based strategies, which require multiple LLM calls (often ~10 per prompt-question pair) to identify the most consistent answer as a pseudo ground truth. This leads to significant computational overhead.
>
> 2. Ambiguity in optimization directions: Other approaches like PromptWizard and SPO depend on LLM-generated pseudo-labels or explicit task specifications. These can be unreliable or unavailable, especially in data-scarce scenarios, and may introduce ambiguity in the optimization direction.
>
> **How does meta-reasoning address these issues?**
>
> In each iteration, meta-reasoning requires only **one additional LLM call** to generate or update task-specific strategies and evaluation criteria. These strategies are incorporated into refined prompts, and the evaluation also focuses on the reasoning process rather than the correctness of the final answer. This shifts the focus from matching a final answer "ground truth" to assessing the problem-solving process, which is more robust than methods relying on weak pseudo-labels.
>
> In our internal evaluation (phase 4), only 3 LLM calls are needed (1 call to run current best prompt, 1 call to run newly optimized prompt, 1 call to compare which prompt is better). This is more efficient than consistency-based methods, which typically require ~10 calls.
>
> In summary, meta-reasoning directly addresses the key limitations of prior self-supervised work by (1) reducing computational cost, and (2) providing a more robust and guided optimization process.
>
>
> ## Clarification on meta-reasoning and reasoning
>
> **Meta-reasoning vs. reasoning:**
> - Meta-reasoning refers to reasoning about the reasoning process itself (i.e., thinking of, reflecting on, and refining strategies for how a problem should be approached). It is not the act of solving the task, but the act of deciding how to solve it. For example, before solving a math problem, you may think of "I shall break the problem into smaller parts".
>
> - Reasoning is actually doing the problem-solving. For example, carrying out the calculations and producing the final answer. It uses the strategies proposed during meta-reasoning.
>
> *Q: Is this mainly referring to phase 1 in your approach?*
>
> Yes, meta-reasoning corresponds to Phase 1 of our framework. In this phase, the model does not try to answer the problem directly. Instead, it analyzes the problem, generates and reflects on potential solution strategies such as decomposition, verification or others.
>
> *Q: What makes this "meta"?*
>
> It is "meta" because the model is not reasoning about the task content (e.g., solving a specific math expression), but reasoning about the strategy it should use (e.g., whether to break the problem into sub-problems, how to verify intermediate steps). This reflection changes how the model thinks through this type of problems rather than giving a final answer to a question.
>
> *Q: Which part of the reasoning is improved?*
>
> The strategies generated in Phase 1 (meta-reasoning) are integrated into the refined prompt in Phase 3. These strategies provide clear guidance on how to approach and solve problems, such as breaking down complex tasks or verifying steps. When the model uses these improved prompts (e.g., Phase 4 execute on validation examples or evaluate on a specific task), it follows a better reasoning process to get the final answer and leads to better overall performance.

---

> ### Author Response · Authors · 2025-11-21
> **Response to Reviewer XDMT (2/4)**
>
> ## Comparison with prior prompt optimization works
>
> Thank you for your suggestion. Below, we clarify which components of our 5-phase optimization loop are newly proposed in MR.PEA and which represent significant improvements over prior self-supervised methods. For reference, supervised methods typically only include prompt refinement and evaluation phases, with prompt refinement based on top-performing prompts and evaluation performed against a labeled training set.
>
> | Phase                        | GLaPE                | PromptWizard                | SPO                        | MR.PEA (Ours)                |
> |------------------------------|----------------------|-----------------------------|----------------------------|------------------------------|
> | 1. Knowledge curation        | ✖️                   | ✖️                          | ✖️                         | ✔️                 |
> | 2. Example generation        | ✖️                   | ✖️                          | ✖️                         | ✔️                  |
> | 3. Prompt refinement         | Consistency score    | Feedback, hand-crafted thinking styles | Top prompt's output      | Feedback, LLM meta-reasoned strategies |
> | 4. Evaluation                | Consistency-based GT | LLM-generated GT            | Manual requirements        | LLM meta-reasoned task-specific criteria|
> | 5. Dynamic prompt management | ✖️                   | ✖️                          | ✖️                         | ✔️            |
>
> Legend:
> ✔️ = included in method; ✖️ = not included; GT = ground truth.
>
> As shown, MR.PEA introduces new phases (knowledge curation, example generation, dynamic prompt management) and enhances prompt refinement and evaluation with meta-reasoning strategies and task-specific criteria.
>
> ## Experimental setting on supervised methods
>
> For supervised methods, we directly used the optimized prompts released by the original authors. These prompts were optimized using GPT-4. According to [OpenAI's evaluation results](https://github.com/openai/simple-evals), GPT-4.1-nano achieves comparable performance to GPT-4. We re-ran these optimized prompts with GPT-4.1-nano for our reported results. It is important to note that these supervised methods may have used part of our test data during their prompt optimization process. This could give these methods an additional advantage in our evaluation.
>
> ## Why MR.PEA is low-cost?
>
> *Q: Since MR.PEA introduces a new meta-reasoning centric module compared to prior work, I'm unsure why MR.PEA is cost-effective compared to the baseline methods. Could you please further clarify? Why are fewer tokens needed comparing to the baselines?*
>
> MR.PEA achieves cost-effectiveness by balancing input and output token usage, resulting in the lowest overall cost according to Azure OpenAI pricing (\\$0.10 per 1M input tokens, \$0.40 per 1M output tokens). As shown in the table below (cost on GSM8K), MR.PEA uses a similar number of API calls as SPO, but fewer tokens overall than most baselines.
>
> | Method        | API Calls | Input Tokens | Output Tokens | Price Cost ($USD) |
> |---------------|-----------|--------------|---------------|-------------------|
> | GLaPE         | 1660      | 322,747      | 198,385       | 0.112             |
> | SPO           | **55**    | 67,946       | **10,744**    | _0.011_           |
> | PromptWizard  | 154       | **39,385**   | 46,671        | 0.023             |
> | MR.PEA (Ours) | _57_      | _48,519_     | _12,173_      | **0.010**         |
>
> Three main designs contribute to MR.PEA’s efficiency:
>
> 1. The phase 1 meta-reasoning knowledge generation or update occurs **at most once per iteration**. If the agent determines the knowledge is sufficient and no update is performed for several consecutive iterations, we skip some rounds and perform periodic re-checks instead.
>
> 2. We use structured outputs in the system design and carefully limit the number of strategies generated in phase 1, which helps control output token length. For example, on GSM8K, Phase 1 typically uses around 1000 input tokens and 250 output tokens when updated, or only about 25 output tokens if no update is needed.
>
> 3. Early stopping is applied when appropriate, reducing the number of iterations and further saving tokens.
>
> For a more detailed breakdown of token consumption per phase, please refer to our response to reviewer bqab. We hope this clarifies how MR.PEA achieves cost-effectiveness compared to prior methods.

---

> ### Author Response · Authors · 2025-11-22
> **Response to Reviewer XDMT (3/4)**
>
> ## Control of prompt length
>
> We do not explicitly constrain prompt lengths during optimization. Instead, we regulate the number of meta-reasoning strategies in phase 1, and set the maximum output token limit to 1024. This ensures that prompts remain concise and focused, without being extremely long.
>
> *Q: In the example in Figure 2, it seems that the prompt generated by MR.PEA is significantly longer than the other two, which could give this method some advantages.*
>
> We clarify that MR.PEA does **not** consistently generate longer prompts than other self-supervised methods. We report both the accuracy and the corresponding prompt length (in tokens) for GSM8K and selected BBH tasks. In the table below, the highest accuracy and minimal prompt length for each task are highlighted in bold.
>
> | Task Accuracy/Prompt Length      | SPO    | PromptWizard | MR.PEA (Ours) |
> |------------------------|--------|--------------|---------------|
> | GSM8K                  | 91.13  | 86.88        | **92.34**     |
> |                        | **75**   | 466           | 86            |
> | boolean_expressions    | 96.40  | 96.00        | **98.80**     |
> |                        | 175    | 394            | **123**           |
> | disambiguation_qa      | 54.50  | 55.60        | **62.40**     |
> |                        | **132**  | 301            | 144           |
> | navigate               | 76.80  | 67.70        | **92.80**     |
> |                        | 149    | 526            | **118**           |
> | object_counting        | 90.00  | 78.40        | **95.20**     |
> |                        | 358    | 522            | **112**           |
> | sports_understanding   | 72.00  | **88.00**    | 84.00         |
> |                        | **102**  | 327          | 196       |
> | temporal_sequences     | 85.20  | 94.80        | **96.80**     |
> |                        | 182    | 668             | **124**           |
>
> As shown in the table, MR.PEA’s prompt lengths typically fall within 100–200 tokens and are often shorter than those from PromptWizard (always above 300 tokens) and, in several cases, shorter than SPO (the main comparison in Figure 2).
>
> Importantly, longer prompts do not guarantee better performance. PromptWizard’s prompts are consistently the longest but do not yield the best results. MR.PEA achieves strong performance with moderate prompt lengths, demonstrating that prompt quality and structure, rather than length-drive improvements.
>
> ## Comparison between knowledge and final prompt
>
> *Q: If "meta-reasoning knowledge" is referring to strategies like "break down complex problems into smaller, manageable steps", this seems to be part of the final prompt of GSM8k ("Break it into clear, sequential steps..."). How is curating knowledge (the problem solving strategy part) different from writing the final prompt?*
>
> Yes. In our framework, "meta-reasoning knowledge" refers to the curated problem-solving strategies, such as "break down complex problems into smaller, manageable steps." These strategies are generated and refined during the meta-reasoning phase.
>
> The final prompt is constructed by integrating the curated knowledge (problem-solving strategies) with a refined task description and output format. In this way, **the knowledge forms a key subset of the final prompt**, guiding how the model approaches each task instance. Refining the final prompt involves combining these strategies with other essential components (task description and output format) to maximize performance. Note that **the final prompt may not necessarily include all curated knowledge**, but only those most relevant for the task.
>
> For a concrete example, please refer to detailed case studies in Appendix C.1 and C.2.

---

> ### Author Response · Authors · 2025-11-22
> **Response to Reviewer XDMT (4/4)**
>
> ## Comparision with Self-Discover
>
> Thank you for highlighting the relevance of Self-Discover. While both MR.PEA (phase 1) and Self-Discover aim to discover problem-solving strategies for a given task type, there are several key differences:
>
>
> | Feature/Method                | Self-Discover                                      | MR.PEA (Ours)                                  |
> |-------------------------------|----------------------------------------------------|------------------------------------------------|
> | Strategy Discovery            | Manually crafted reasoning modules +  demos   |  Automatically discovers strategies from LLM    |
> | Process             | One-time, fixed structure        | Iterative, allows reflection and adjustment    |
> | Flexibility            | Manual effort if new modules required                 | Adapts automatically, no manual intervention    |
> | Generalization (given one example)    | May be task-specific, limited generalization        | More generalizable, avoids question-specific info |
> | Output                 | Reasoning structure only                           | Final optimized prompt for the task             |
>
> **Self-Discover** relies on 39 manually crafted reasoning modules and reasoning structure demonstrations. While this approach can be effective, the reasoning modules for selection is limited, which limits flexibility. If a new reasoning module is needed, users need to craft it manually. In addition, without good demonstrations, the generated reasoning structures by Self-Discovery tends to be less-effective. Especially in cases with only one example, the generated structure often incorporates details specific to that sample, which limits its ability to generalize to other task instances.
>
> **MR.PEA**, on the other hand, automatically discovers actionable, reusable strategies from the LLM’s internal knowledge. The meta-reasoning process is iterative, allowing the system to reflect and adjust strategies if earlier generation are suboptimal. With only one sample, MR.PEA’s strategies remain generalizable because they do not incorporate question-specific information. Importantly, MR.PEA is designed not just to generate strategies, but to produce the final optimized prompt for the task.
>
>
> Below, we show an example of the reasoning structure generated by Self-Discover with demonstration, using the author's examples from Figure 2 for GSM8K. This approach produces a multi-step reasoning plan, which in this case consists of 7 steps. For example:
>
> ```json
>     "Step 1": {
>       "Identify key information": "Extract the number of games Jenny played against each friend, the number of wins for each, and any relevant percentages or ratios provided."
>     },
>     "Step 2": {
>       "Define variables": "Assign variables to unknown quantities, such as total games played with each friend, total wins, and total games played overall."
>     }
>     // ...additional steps...
> ```
>
> In contrast, MR.PEA's reasoning strategies in the final prompt is more concise and general:
> ```
> Break it into clear, sequential steps, performing all intermediate calculations explicitly with proper notation. Verify each step for accuracy. Use consistent terminology and formatting. Summarize the reasoning process briefly, explicitly connect each step to the question, and conclude with the final answer, ensuring clarity, correctness, and completeness.
> ```
>
> To further illustrate the practical impact, we compare the performance of MR.PEA and Self-Discover (with and without demonstration) on GSM8K (500 samples) and BBH Navigate tasks:
>
> | Task         | Self-Discover w/o Demo | Self-Discover w/ Demo | MR.PEA (Ours) |
> |--------------|-----------------------|----------------------|---------------|
> | GSM8K  | 57.0  | 86.4                 | **92.8**      |
> | BBH Navigate | 51.2                  | 68.8                 | **92.8**      |

---

### Official Review · Reviewer_UQ6n · 2025-10-31

**Soundness:** 2
**Presentation:** 3
**Contribution:** 3
**Rating:** 4
**Confidence:** 2

**Summary:**

The paper proposes MR.PEA, a self-supervised framework for prompt optimization that uses meta-reasoning to build task-specific strategies and evaluation criteria from minimal input. It iteratively refines prompts through knowledge curation, example generation, evaluation, and ranking. Results on GSM8K and Big-Bench Hard show clear gains over existing methods at very low cost.

**Strengths:**

- Conceptually interesting meta-reasoning approach with minimal supervision and input requirements.
- Well-structured and modular iterative framework.
- Strong empirical results especially on Math tasks.
- Clear ablation results showing each component’s contribution.
- Low cost and scalability demonstrated.

**Weaknesses:**

- Internal decision logic remains heuristic and not quantitatively validated.
- Evaluation limited to GPT-4.1-nano with temperature of 0.2, leaving generalization and robustness untested.
- Comparisons (Table 1) may be partially unfair, as MR.PEA can access external web information while most baselines cannot.

**Questions:**

- Have you evaluated MR.PEA’s generalization across different model families or sizes beyond GPT-4.1-nano?
- Have you analyzed how sensitive MR.PEA’s prompt generation is to variations in internal heuristic choices and decoding temperature?
- To what extent can the reported performance gains of MR.PEA over baselines be attributed to its access to external web information?

---

> ### Author Response · Authors · 2025-11-21
> **Response to Reviewer UQ6n (1/2)**
>
> We thank the reviewer UQ6n for the thoughtful feedback and for recognizing the originality of our low-cost meta-reasoning approach with the minimal supervision and input requirements, strong empirical results, clear ablation studies. Below, we address the concerns.
>
> ## Generalization across different model families and sizes
>
> We conducted additional experiments with models of varying sizes within the same family (GPT-4.1-nano, GPT-4.1, GPT-4.1-mini) and across different model families (DeepSeek-V3, Llama-3.3-70B-Instruct), using the BBH geometric shapes task. We considered two prompt optimization scenarios: (1) using prompts optimized by GPT-4.1-nano (as in the main paper) to test prompt transferability across models, and (2) using prompts optimized by each target model to assess the framework’s adaptability. The results are shown below:
>
> ### Different model families
> | Model                        | DeepSeek-V3 | Llama-3.3-70B-Instruct|
> |------------------------------|-------------|-----------------------|
> | CoT baseline                 | 70.4        | 60.0                  |
> | MR.PEA (GPT-4.1-nano prompt) | 68.8        | **72.4**              |
> | MR.PEA (self-optimized)      | **70.8**    | 63.6                  |
>
> Llama-3.3-70B-Instruct, an instruction-tuned model, shows a substantial improvement over the CoT baseline when using prompts optimized by GPT-4.1-nano (from 60.0 to 72.4). This suggests that MR.PEA is effective at enhancing the reasoning performance of instruction-following models. In contrast, models like DeepSeek-V3, which are already specialized for complex reasoning, have less headroom for further gains through prompt optimization.
>
> ### Same model family, different sizes
> | Model                        | GPT-4.1-mini | GPT-4.1 |
> |------------------------------|--------------|---------|
> | CoT baseline                 | 62.8         | 74.0    |
> | MR.PEA (GPT-4.1-nano prompt) | 71.2         | 73.6    |
> | MR.PEA (self-optimized)      | **72.8**     | **74.4**|
>
> Within the GPT-4.1 family, MR.PEA consistently outperforms the CoT baseline across different model sizes. Prompts optimized by GPT-4.1-nano transfer well to GPT-4.1-mini, and self-optimized prompts yield the best results for both models. The performance boost is more significant for smaller models.
>
>
> ## Evaluation on different decoding temperatures
> To assess the sensitivity of optimized prompts to decoding temperature, we conducted experiments on GSM8K (500 samples) and BBH Navigate tasks. Decoding temperature controls the randomness of model outputs: lower temperatures yield more deterministic responses, while higher temperatures introduce greater variability. We evaluated performance at temperatures 0, 0.1, 0.2, 0.3, 0.4, and 1.0.
>
> | Temperature | 0    | 0.1  | 0.2  | 0.3  | 0.4  | 1    |
> |-------------|------|------|------|------|------|------|
> | GSM8K       | 92.8 | 93.4 | 92.8 | 93.0 | 92.4 | 92.4 |
> | Navigate    | 93.6 | 93.6 | 92.8 | 93.2 | 94.0 | 91.4 |
>
> We observe that MR.PEA’s performance remains relatively stable across a range of low to moderate temperatures. Notably, at temperature 1.0, the accuracy on BBH Navigate drops, primarily because the model is less likely to follow the required answer format.

---

> ### Author Response · Authors · 2025-11-21
> **Response to Reviewer UQ6n (2/2)**
>
> ## Attribution of gains to web access
> We appreciate the reviewer UQ6n’s concern regarding the potential impact of web access on the fairness of comparisons. To clarify, in our reported results, only three BBH tasks—movie_recommendation, ruin_names, and snarks actually benefit from web search. For all other tasks, MR.PEA relies solely on its internal knowledge.
>
> To directly address this concern, we report MR.PEA’s performance on BBH and GSM8K sub-areas with web search completely disabled (see table below). This is a non-web search version of Table 1 for a fair comparison. Results demonstrates that MR.PEA still outperforms all baselines across nearly all categories, indicating that the majority of the performance gains are not attributable to web access. The framework remains effective even when restricted to internal model knowledge.
>
> | Model      | OPRO |GPO    |GLaPE | SPO | PromptWizard | MR.PEA w/o web (Ours) |
> |------------|-----|-----|-----|-----|-----|-----|
> | GSM8K      | 90.98|90.07|81.50 | 91.13   | 86.88  |  **92.34**  |
> | Common Sense|  62.06|52.16| 54.64 | 67.60   | 63.23  |  **68.91**  |
> | Language   |  53.08| 55.70 | 50.40 | **66.62**   | 64.65  |  65.32  |
> | Logic      |  65.20|64.96| 59.04 | 83.36   | 74.32  |  **85.84**  |
> | Math       |  63.28| 70.72| 59.92 | 69.20   | 67.44  |  **84.00**  |
> | Spatial/Seq./Attr. | 60.33| 66.95|  62.53 | 79.25   | 77.53 |  **87.23**  |
> | **Average**   | 65.02| 63.02|  58.06 | 73.68   | 69.81  |  **78.46**  |
>
>
>
> ## Concerns on internal decision logic
>
> In MR.PEA, the agent’s internal decision-making such as whether and how to update its knowledge is not governed by fixed, hand-crafted heuristics. Instead, these decisions emerge dynamically from the LLM’s meta-reasoning process, which is guided by the current task context rather than explicit rules or thresholds.
>
> To provide quantitative insight, we report the number of knowledge update steps for several tasks: for GSM8K, the agent updated its knowledge once; for temporal_sequences, four times; and for geometric_shapes and object_counting, five times. Across tasks, the number of updates typically ranges from one to ten. This variability reflects the agent’s autonomous and adaptive assessment of when further refinement is needed, rather than any pre-set parameter.
>
> We believe that in our system the soundness of the internal decision logic is best validated by the overall effectiveness of the process, rather than by the optimality of each individual decision. As shown in Table 2 (ablation study on each component), disabling meta-reasoning leads to a performance drop ranging from 1.6% to 18.2%.
>
> If the reviewer UQ6n is referring to a different aspect of "internal decision logic" or "heuristic choices," we would appreciate further clarification so we can address the concern more precisely.

---

### Official Review · Reviewer_bqab · 2025-11-01

**Soundness:** 3
**Presentation:** 2
**Contribution:** 3
**Rating:** 8
**Confidence:** 4

**Summary:**

This paper introduces the MR.PEA, a novel self-supervised prompt optimization framework. The primary goal of MR.PEA is to overcome the limitations of current LLM prompt optimization methods, namely their reliance on costly labeled data and the optimization ambiguity or high computational cost associated with traditional self-supervised approaches. The core mechanism of MR.PEA is meta-reasoning, which allows the agent to iteratively build and refine task-specific knowledge using only minimal input. This approach aims to achieve efficient and adaptive prompt optimization in a self-supervised manner.

**Strengths:**

1. The use of a "meta-reasoning" mechanism for prompt engineering presents an innovative paradigm distinct from conventional search-based or LLM-generation/filtering optimization methods. This mechanism promises to make the optimization process more interpretable and directed.

2. The framework's emphasis on "self-supervision" and "minimal input" offers significant practical value by substantially reducing the cost and complexity of deploying LLM applications, especially in domains where large amounts of labeled data are scarce.

3. MR.PEA iteratively distills general strategies and principles. This means the output is not just an optimal prompt, but a reusable "prompt engineering knowledge base" applicable to similar tasks.

4. Prompt engineering is a critical bottleneck for the practical deployment of LLM-based systems. Proposing an automated, low-cost optimization framework is highly relevant and timely in the current research landscape.

**Weaknesses:**

1. While data cost is reduced, the meta-reasoning mechanism itself typically involves multiple, complex LLM calls and knowledge structuring steps. The paper needs to clearly quantify and compare the overhead of MR.PEA in terms of total token consumption and latency against relevant baselines.

2. Given the inherent uncertainty in LLM generation, the robustness of MR.PEA's iterative optimization process is vital. The authors should provide evidence of performance stability and generalization capability across diverse task domains (e.g., classification, summarization, code generation) and when using different LLM backbones.

**Questions:**

See Weakness

---

> ### Author Response · Authors · 2025-11-20
> **Response to Reviewer bqab (1/2)**
>
> We thank reviewer bqab for the thoughtful and constructive feedback. We greatly appreciate your recognition of MR.PEA’s novelty, practical value, and its potential to advance low-cost prompt optimization. We appreciate the reviewer bqab’s concerns regarding the computational overhead of MR.PEA and the need to demonstrate its generalization across diverse tasks and LLM backbones. We address these points in detail below.
>
> ## Quantifying token consumption and latency
>
> We provide a detailed cost comparison of self-supervised methods on GSM8K in the table below. All methods follow the authors' default settings, with inputs adapted to match our method (one data sample without answers). All experiments utilize GPT-4.1-nano as the base LLM, and the price cost is estimated based on Azure OpenAI's pricing (\\$0.10 per 1M input tokens and \$0.40 per 1M output tokens).
>
> | Method        | API Calls | Input Tokens | Output Tokens | Price Cost ($USD) | Time (minutes) |
> |---------------|-----------|--------------|---------------|-------------------|----------------|
> | GLaPE         | 1660      | 322,747      | 198,385       | 0.112             | ~90            |
> | SPO           | **55**    | 67,946       | **10,744**    | _0.011_           | **~1.5**       |
> | PromptWizard  | 154       | **39,385**   | 46,671        | 0.023             | ~12            |
> | MR.PEA (Ours) | _57_      | _48,519_     | _12,173_      | **0.010**         | _~2_           |
>
>
> To further clarify MR.PEA’s overhead, we break down the approximate token usage for each phase of the pipeline on GSM8K. In the results above, MR.PEA updated the knowledge only once. If knowledge is updated at every iteration, the total token and time cost increases moderately, as shown below.
>
> | Phase        | Input tokens | Output tokens |
> |---------------|-----------|--------------|
> | Phase 1 - knowledge curation    | ~1000      | ~250 if update; ~25 if no update     |
> | Phase 2 - example generation    | ~1500      | ~250        |
> | Phase 3 - prompt refinement    | ~1300      | ~150      |
> | Phase 4 - evaluation execution    | ~180*2      | ~330*2      |
> | Phase 4 - evaluation comparison   | ~1600     | ~130      |
>
> | Total        | Input tokens | Output tokens |
> |---------------|-----------|--------------|
> | Update knowledge once   | ~54,600     | ~12,300      |
> |Update knowledge every time   | ~57,600     | ~14,400      |
>
> In practice, updating knowledge once typically takes 2–2.5 minutes, while updating at every iteration would require 3 more calls and 2.5–3 minutes in total. This is because our implementation includes a mechanism to skip unnecessary updates: if the knowledge has already been sufficiently reflected in recent iterations (i.e., no changes in knowledge), we automatically skip some rounds of phase 1. However, after a few skipped rounds, we periodically re-enable phase 1 to ensure that any new knowledge or changes are still captured in the optimization process.
>
> Additionally, if there are early exits, the cost can be further reduced. For example, in the object_counting task, the process finished early at iteration 5, taking only about 1 minute, with 21,218 input tokens and 5,764 output tokens.

---

> ### Author Response · Authors · 2025-11-21
> **Response to Reviewer bqab (2/2)**
>
> ## Generalization across different LLMs
>
> To evaluate the robustness and generalization of MR.PEA, we conducted additional experiments with models of different sizes within the same family (GPT-4.1-nano, GPT-4.1, GPT-4.1-mini) as well as across different model families (DeepSeek-V3, Llama-3.3-70B-Instruct), using the BBH geometric shapes task. We considered two prompt optimization scenarios: (1) using prompts optimized by GPT-4.1-nano (as in the main paper) to test prompt transferability across models, and (2) using prompts optimized by each target model itself to assess the framework’s adaptability. The results are shown below:
>
> | Model                        | GPT-4.1-mini | GPT-4.1 | DeepSeek-V3 | Llama-3.3-70B-Instruct |
> |------------------------------|--------------|---------|-------------|-----------------------|
> | CoT baseline                 | 62.8         | 74.0    | 70.4        | 60.0                  |
> | MR.PEA (GPT-4.1-nano prompt) | 71.2         | 73.6    | 68.8        | **72.4**                  |
> | MR.PEA (self-optimized)      | **72.8**     | **74.4**    | **70.8**        | 63.6                  |
>
> Across all tested models, MR.PEA consistently matches or outperforms the CoT baseline with the prompt either optimized by GPT-4.1-nano or by each model itself. This highlights both the generality and effectiveness of our approach. Notably, prompts optimized by GPT-4.1-nano transfer well to other models, yielding substantial improvements, particularly for smaller or less reasoning-focused models, such as Llama-3.3-70B-Instruct (from 60.0 to 72.4) and GPT-4.1-mini (from 62.8 to 71.2). Usually, when each model uses its own optimized prompt, MR.PEA achieves the best results for that model, further demonstrating the adaptability of our framework.
>
> We also observe that instruction-tuned models (such as the GPT series and Llama-3.3-70B-Instruct) benefit most from prompt optimization, while reasoning-focused models (like DeepSeek-V3) see relatively smaller improvements. This suggests that MR.PEA is effective across model types, with especially strong gains for models that are better at following instructions but have relatively weaker inherent reasoning abilities.
>
>
> ## Performance on classification, summarization and code generation
>
> We evaluated MR.PEA across three diverse task domains: classification, summarization, and code generation. All experiments were conducted using GPT-4.1-nano for consistency.
>
> - **Classification (IMDB reviews):**
>   We randomly selected 300 samples from the IMDB review dataset ([link](https://www.kaggle.com/datasets/lakshmi25npathi/imdb-dataset-of-50k-movie-reviews)).
>   - Manual prompt: 94.00% accuracy
>   - MR.PEA optimized prompt: 94.60% accuracy
>
> - **Summarization (CNN/DailyMail):**
>   We randomly selected 300 samples from the CNN/DailyMail dataset ([link](https://huggingface.co/datasets/abisee/cnn_dailymail)).
>   - Manual prompt (ROUGE-1/2/L): 0.288 / 0.089 / 0.175
>   - MR.PEA optimized prompt (ROUGE-1/2/L): 0.278 / 0.076 / 0.164
>
> MR.PEA’s ROUGE scores are slightly lower than the manual prompt. This is mainly because MR.PEA tends to generate longer and more detailed highlights, often including background information and analysis. Since we used "output highlight points for the article" as the task description, MR.PEA was not able to infer the preferred output format (e.g., concise event summaries). For summarization tasks, explicitly specifying the desired output style or format in the prompt would likely help.
>
> - **Code Generation (HumanEval):**
>
> We evaluated MR.PEA on the HumanEval benchmark ([link](https://github.com/openai/human-eval)), comparing manual prompts with MR.PEA-optimized prompts. The results below report pass@k (k=1,2,5) along with their standard deviation:
>
> | Metric   | Manual Prompt      | MR.PEA Optimized Prompt |
> |----------|-------------------|------------------------|
> | pass@1   | 0.873 (± 0.297)   | **0.888 (± 0.269)**    |
> | pass@2   | 0.901 (± 0.270)   | **0.922 (± 0.245)**    |
> | pass@5   | 0.933 (± 0.250)   | **0.945 (± 0.228)**    |
>
> MR.PEA consistently manual prompts across all pass@k metrics. Additionally, MR.PEA achieves lower standard deviation, meaning its results are more consistent and stable across different samples.

---

### Official Review · Reviewer_ak1k · 2025-11-03

**Soundness:** 3
**Presentation:** 4
**Contribution:** 2
**Rating:** 4
**Confidence:** 5

**Summary:**

This paper introduces a simple self-supervised prompt optimization approach MR.PEA, that uses minimal inputs to curate some strategies and evaluation criteria, generates different validation examples, refines prompts using the knowledge, and conducts criteria based pairwise comparisons while maintaining a ranked prompt pool with explore/exploit scoring. Additionally, it also leverages web search for extracting related knowledge to improve the prompt. Overall, the methodology is incremental in novelty for prompt optimization.

**Strengths:**

1. Mr. PEA seems to be a well engineered system, using a well orchestrated and meta prompted pipeline it can improve performance. The five phase loop is clearly specified with pseudocode and control parameters.
2. Mr. PEA can improve the prompt with only only example as the seed.
3. The paper writing is really easy to follow.

**Weaknesses:**

1. The proposed methodology is quite incremental to the ongoing works in this domain. The whole system is a simple loop of orchestrated LLM calls with very well engineered prompts. That significantly lowers the novelty of the proposed approach.
 2. My primary concern lies in allowing the LLM to search web for related information. As it searches it should ofcourse get significantly more information regarding the task. It is not really surprising that it is one of the primary aspects of improved performance. Also, for optimization, it can search upto 20 times in the web, that is a significant amount of additional information which definitely should improve performance.
 3. I could not find the details about it, but web searching for a task like GSM8K or BBH tasks significantly increases the risk of seeing more data as well as gathering more information. A simple web search with one of the examples, already bring up all the examples in the web from that dataset (BBH). I think with web searches, prompt optimization needs to be done on more sophisticated tasks to showcase the performance benefits.
 4. Baselines suggestions: comparison against APO, PE2, TextGrad would be interesting to see.

**Questions:**

Please see the weaknesses.

---

> ### Author Response · Authors · 2025-11-20
> **Response to Reviewer ak1k (1/3)**
>
> We thank Reviewer ak1k for the constructive feedback and for recognizing the strengths of MR.PEA, specifically regarding the well-engineered system, the effectiveness of our approach using only a single seed example (without answers), and the presentation of our writing. We address the concerns regarding novelty, web search, and baselines below.
>
> ## Novelty of the system
> While MR.PEA leverages LLMs as optimizers, our key contribution is a **framework specifically designed for prompt optimization under extreme data scarcity—using only a single data sample without ground truth**. To our knowledge, existing self-supervised prompt optimization methods are not effective in this setting.
>
> MR.PEA addresses two core limitations of prior work:
> 1. Optimization ambiguity: Existing methods (e.g., PromptWizard, SPO) often rely on LLM-generated pseudo-labels or require explicit task specifications, which can be unreliable or unavailable in data-scarce scenarios. In contrast, MR.PEA leverages meta-reasoning to autonomously generate and refine both problem-solving strategies and evaluation criteria. This enables more directed and robust internal validation by focusing on the reasoning process rather than just the final answer. Our ablation study (Table 2) highlights the critical role of these components.
>
> 2. Efficiency (LLM calls): Approaches such as GLaPE rely on computationally intensive consistency-based validation (~10 calls) during optimization. MR.PEA instead uses meta-reasoned evaluation criteria to validate. In each iteration, the evaluation only needs 3 calls (1 call to run current best prompt, 1 call to run newly optimized prompt, 1 call to compare which prompt is better).
>
> To make MR.PEA more robust and efficient in the single data sample scenario, we further employ two key strategies. First, to mitigate overfitting, MR.PEA generates a new and diverse example at each iteration, which encourages generalization beyond the initial data point. Second, we carefully design prompts for each stage of the pipeline (e.g., by limiting the number of strategies generated per iteration and enforcing structured outputs). These design choices enable MR.PEA to optimize prompts reliably, while also controlling token usage.
>
> We note that iterative loops are fundamental to almost all prompt optimization frameworks. Compared with existing works, MR.PEA lowers the barrier for prompt optimization (i.e., data and computation cost), which makes it practical for real-world usage.

---

> ### Author Response · Authors · 2025-11-20
> **Response to Reviewer ak1k  (2/3)**
>
> ## Concerns on web search
>
> We thank the reviewer ak1k for raising this concern regarding the potential risk of data contamination and unfair advantage via web search. We agree that if a model could access the test datasets online during optimization, it would indeed invalidate the results.
>
> ### Clarification and correction
> We would like to clarify a typo in our original manuscript: MR.PEA is not allowed ''up to 2 searches per iteration'', but rather **''up to 2 web searches throughout the entire optimization process''**. We have updated the paper accordingly.
>
> Web search is strictly limited and is not a primary source of information for MR.PEA. In practice, web search is expensive and may be ineffective. The agent is encouraged to rely on meta-reasoning and only use web search as a last resort. Our design ensures that MR.PEA cannot over-rely on external information.
>
> In our reported results, **only three tasks (movie_recommendation, ruin_names, and snarks from BBH) actually benefit from web search.** For the other tasks, the agent relied solely on its internal knowledge. Importantly, the initial task prompt never includes dataset names (e.g., GSM8K, BBH), so the agent cannot intentionally search for dataset-specific solutions. This minimizes the risk of data leakage.
>
> Currently, our web search module is naively implemented. Agent-generated queries are often suboptimal and may fail to retrieve useful information during the search process. Take BBH_ruin_names for example, it searched "examples of humorous one-character edits creating valid words" at the first attempt, but no examples returned from the search engine. Later it searched "methods for generating humorous one-character word edits with valid words" and successfully incorporate search results in its knowledge. More details are in Appendix C.2. In some cases, web search even introduces noise and can hurt performance (e.g., for web_of_lies, accuracy dropped from 86.4 to 83.2 with web search).
>
> ### Performance without web search
>
> To directly address the reviewer ak1k’s concern, we report MR.PEA’s performance (GSM8K and BBH sub-areas) when web search is completely disabled. Full results are in Table 4. As shown below, MR.PEA maintains strong results and consistently outperforms or matches existing self-supervised baselines.
>
> | Model      | GLaPE | SPO | PromptWizard | MR.PEA w/o web (Ours)  |
> |------------|----------|----------|----------|----------|
> | GSM8K      | 81.50 | 91.13   | 86.88  |  **92.34**  |
> | Common Sense|  54.64 | 67.60   | 63.23  |  **68.91**  |
> | Language   |  50.40 | **66.62**   | 64.65  |  65.32  |
> | Logic      |  59.04 | 83.36   | 74.32  |  **85.84**  |
> | Math       |  59.92 | 69.20   | 67.44  |  **84.00**  |
> | Spatial/Seq./Attr. |  62.53 | 79.25   | 77.53 |  **87.23**  |
>
> Average performance on GSM8K and BBH:
> | Model      | GLaPE | SPO | PromptWizard | MR.PEA w/o web (Ours)  |
> |------------|----------|----------|----------|----------|
> | Accuracy   |  58.06 | 73.68   | 69.81  |  **78.46**  |
>
>
> These results demonstrate that the primary strength of MR.PEA lies in its meta-reasoning, not in external data. Web search is an optional and imperfect tool for knowledge curation, which can be replaced or integrated with more reliable data sources in the future.

---

> ### Author Response · Authors · 2025-11-20
> **Response to Reviewer ak1k  (3/3)**
>
> ## Comparioson with other supervised prompt optimization baselines
>
> In response to the reviewer ak1k's suggestion, we conducted a comparison of MR.PEA with three supervised prompt optimization baselines: APO, PE2, and TextGrad. We evaluated all methods on GSM8K and representative tasks from the BBH benchmark (at least one task from each sub-area).
>
> For APO and PE2, we used the optimized prompts provided in the PE2 paper, where the authors utilized GPT-4/GPT-4-Turbo for optimization. According to [OpenAI's evaluation results](https://github.com/openai/simple-evals), GPT-4.1-nano achieves comparable performance to GPT-4/GPT-4-Turbo, so we directly adopted their optimized prompts for comparison. For TextGrad, we re-ran the optimization using the authors' default settings with GPT-4.1-nano. All evaluations were performed using GPT-4.1-nano on the the full test set (GSM8K) and full dataset (BBH) for each task.
>
>
> | Task         | APO | PE2 | TextGrad | MR.PEA (Ours)  |
> |--------------|----------|----------|----------|----------|
> | GSM8K                 |  91.21 |  90.67   | 90.14   |  **92.34**  |
> | boolean_expressions   | 86.80    | 94.00    | 97.60   | **98.80**    |
> | disambiguation_qa     | 50.00    |  27.60    | 59.20   | **62.40**   |
> | object_counting       |  86.40    | 50.00    |  **95.60**    |  95.20     |
> | sports_understanding  | 67.60    | 54.00    | 79.60     | **84.00**    |
> | temporal_sequences    |  69.20   |  61.20    |  93.20     |  **96.80**   |
> | navigate              |  66.40    | 88.80    |  92.00   |  **92.80**    |
>
> As shown in the table, MR.PEA consistently outperforms APO and PE2 across all tasks, and matches or exceeds the performance of TextGrad. Importantly, MR.PEA achieves these results with only about 60 optimization calls per task, whereas TextGrad typically requires hundreds or even thousands of calls, not to mention the additional data preparation. This highlights MR.PEA’s core contribution: enabling efficient and effective prompt optimization under extreme data scarcity.

---

> > ### Comment · Reviewer_ak1k · 2025-11-27
> > **Response to Author Rebuttal**
> >
> > I sincerely thank the authors for conducting the comparison against other prompt optimization approaches. I'll increase my soundness score, therefore. I will keep my final rating due to my concerns about the contribution and overall novelty beyond the framework design.

---

> > > ### Author Response · Authors · 2025-11-28
> > >
> > > Dear Reviewer ak1k,
> > >
> > > We sincerely appreciate your careful re-evaluation and the increase in the soundness score. To briefly clarify the contribution:
> > >
> > > (1) **A new problem setting**: We study prompt optimization under extreme supervision scarcity (single example, no labels), a setting where existing methods do not operate effectively.
> > >
> > > (2) **A new optimization signal**: Our core contribution is introducing meta-reasoning knowledge as a self-supervised objective, allowing the model to strategically improve prompts by itself rather than relying on accuracy, consistency or manual-defined signals. This constitutes a conceptual shift beyond prior frameworks.
> > >
> > > (3) **A practical contribution**: MR.PEA achieves both higher performance and lower cost than existing approaches.
> > >
> > > We fully respect the reviewer’s perspective, and we hope this clarifies that the contribution extends beyond framework design, offering a new conceptual grounding and enabling an efficient paradigm for label-free prompt optimization.

---

### Author Response · Authors · 2025-12-01
**Summary of the discussion**

We thank all reviewers for their thoughtful feedback. Below we provide a concise summary addressing the main concerns, and note that we have added further clarifications in the main paper and appendix wherever appropriate.

1. Novelty/motivation (ak1k, XDMT):
Our work targets prompt optimization under extreme supervision scarcity (single example, no labels), a setting where prior methods perform poorly. Our core contribution is using meta-reasoning knowledge as a self-supervised optimization signal. This enables directed and efficient optimization without increased cost, as the meta-reasoning step requires only at most one additional LLM call per iteration.

2. Concerns about web search (ak1k, UQ6n):
Only 3 of 23 tasks use web search. We provide non-web results showing MR.PEA remains strong, especially for Math, Spatial/Sequence/Attribute tasks.

3. More baseline comparisons
	- Experimental results against supervised baselines (APO, PE2, TextGrad) (ak1k): MR.PEA achieves mostly better  or comparable performance while using far fewer LLM calls.
	- Component analysis against prior self-supervised methods (XDMT): MR.PEA newly introduces meta-reasoning knowledge curation, example generation, dynamic prompt management, which are not present previously.
	- Against Self-Discover (XDMT): We provide structural comparisons and empirical results showing that the two methods are fundamentally different and that MR.PEA is more effective in the low-data setting.

4. Token/cost efficiency (bqab, XDMT):
We report actual optimization costs (tokens, price, latency). MR.PEA achieves the lowest price cost, and the optimized prompts are not necessarily longer than those of the baselines.

5. Generalization (bqab, UQ6n):
MR.PEA generalizes across tasks (code and classification) and across models (different families and sizes). Prompts optimized with smaller models can transfer to other instruct models.

---

### Meta-Review · Area_Chair_oVA9 · 2026-01-08

**Summary:**

This paper presents a self-supervised prompt optimization framework called Meta-Reasoning Prompt Engineering Agent (MR.PEA). It leverages meta-reasoning to iteratively retrieve and build task-specific knowledge for prompt optimization. Experiments on GSM8K and Big-Bench Hard show that MR.PEA outperforms several baselines.

**Reviewer Concerns:**

1. limited novelty, very incremental work. i don't think the authors well addressed this concern.
2. unfair setup. the proposed method uses additional information from web search. I think the authors addressed it by providing additional ablation studies.
3. more baselines and evaluation. I think the authors addressed it by providing additional experiment results.
4. more explanation of the underlying motivation.   I think the authors partly addressed it.

**Reviewer Scores:**

I don't think the reviewers will change their scores, as they expressed their primary concern is on the novelty of this work.

---

### Decision · Program_Chairs · 2026-01-26

Reject